# TRANSPORT-BASED MEAN FLOWS FOR GENERATIVE MODELING

## ABSTRACT

Flow-matching generative models have emerged as a powerful paradigm for continuous data generation, achieving state-of-the-art results across domains such as images, 3D shapes, and point clouds. Despite their success, these models suffer from slow inference due to the requirement of numerous sequential sampling steps. Recent work has sought to accelerate inference by reducing the number of sampling steps. In particular, Mean Flows offer a one-step generation approach that delivers substantial speedups while retaining strong generative performance. Yet, in many continuous domains, Mean Flows fail to faithfully approximate the behavior of the original multi-step flow-matching process. In this work, we address this limitation by incorporating optimal transport–based sampling strategies into the Mean Flow framework, enabling one-step generators that better preserve the fidelity and diversity of the original multi-step flow process. Experiments on controlled low-dimensional settings and on high-dimensional tasks such as image generation, image-to-image translation, and point cloud generation demonstrate that our approach achieves superior inference accuracy in one-step generative modeling. The code for re- producing all the numerical results is available in the anonymous repository at `https://anonymous.4open.science/r/OT-flow-FE8F/`.

## 1 INTRODUCTION

Flow-based generative models have emerged as a cornerstone of modern generative AI, providing a unifying framework for modeling complex continuous data distributions. The goal is to transform a source distribution (which may be simple, such as a Gaussian, or complex, as in image-to-image translation) into a target data distribution (e.g., natural images). Two prominent frameworks are *diffusion models* and *flow matching (FM)*. Diffusion models formulate generation via a stochastic differential equation (SDE) and learn a score function or denoising function (Sohl-Dickstein et al., 2015; Ho et al., 2020; Song et al., 2021; Dhariwal & Nichol, 2021; Karras et al., 2022), whereas flow matching uses an ordinary differential equation (ODE) and learns a velocity field to continuously transform the source distribution into the target (Lipman et al., 2023; Albergo & Vanden-Eijnden, 2023; Liu et al., 2022). These two methods are closely related: under Gaussian priors and independent couplings between source and target, they can be converted into one another (De Bortoli et al., 2023; Tong et al., 2023b).

A key limitation of both diffusion and classical FM models is that, during inference, one must numerically solve an integration problem, which requires many steps to obtain accurate results (Song et al., 2021; Lipman et al., 2023). To mitigate this issue, several complementary directions have been explored. One large body of work distills a multi-step diffusion model (teacher) into a one-step generator (student) via trajectory or distribution matching objectives, with recent advances including adversarial distillation, consistency models, and $f$-divergence–based approaches (Meng et al., 2023; Song et al., 2023; Sauer et al., 2024; Yin et al., 2024; Xu et al., 2025). Another line of work seeks to train FM or diffusion models with inherently straighter and more cost-efficient trajectories. For instance, Tong et al. (2023a); Kornilov et al. (2024); Pooladian et al. (2023) leverage optimal transport to define the joint sampling between source (Gaussian) and target data, yielding straighter trajectories and improved efficiency both theoretically and empirically (Pooladian et al., 2023; Liu et al., 2022). More recently, Geng et al. (2025) introduces the *MeanFlow* method, which replaces

the instantaneous velocity field with its time-averaged integration as the learning target. This reformulation enables one-step or few-step sampling, significantly accelerating inference.

In this paper, we propose the *Optimal Transport MeanFlow (OT-MF)* method, which unifies the trajectory-straightening principles of optimal transport with the time-averaged formulation of Mean-Flow, yielding straighter and more efficient one-step generative trajectories. Our approach combines OT-based couplings with mean-flow supervision, enabling geometry-aware and efficient one-step generative modeling. Our contributions are summarized as follows:

- **Unified framework.** We introduce OT-MF, a new flow matching method that generalizes conditional flow matching, minibatch OT flow matching, and mean flow approaches under a single formulation.
- **Improved efficiency and accuracy.** In point cloud generation and image translation experiments, OT-MF retains the one-step generation ability of MeanFlow while significantly improving accuracy.
- **Scalable training.** To further enhance training efficiency, we incorporate accelerated OT solvers including linear OT and hierarchical OT. We demonstrate that these extensions preserve the accuracy of OT-MF while reducing computational cost during training.

## 2 BACKGROUND AND RELATED WORK

**Notation Setup and ODE.** Suppose the dataset lies in the space $\mathbb{R}^d$. Let $\mathcal{P}(\mathbb{R}^d)$ denote the set of all probability measures on $\mathbb{R}^d$. We use $\mathbf{p}, \mathbf{q} \in \mathcal{P}(\mathbb{R}^d)$ to denote the source (prior) and target (data) laws, respectively; their densities (when they exist) are denoted $p(x), q(x)$. By default, $\mathbf{p}$ is taken to be the Gaussian law, i.e., $\mathbf{p} = \mathcal{N}(0, I_d)$.

We also define the following ODE system:

$$\begin{cases} \psi : [0,1] \times \mathbb{R}^d \to \mathbb{R}^d, & (t, x_0) \mapsto \psi_t(x_0), \\ v : [0,1] \times \mathbb{R}^d \to \mathbb{R}^d, & (t, x) \mapsto v(t, x) := v_t(x), \\ d\psi_t(x_0) = v_t(\psi_t(x_0)) \, dt & \text{(flow ODE)}, \\ \psi_0(x_0) = x_0 & \text{(initial condition)}. \end{cases} \tag{1}$$

Here, $v_t$ is called the **time-dependent vector/velocity field**, and the solution $\psi$ is referred to as the **time-dependent flow**. We say that the velocity field $v$ generates a probability path $(\mathbf{p}_t)_{t \in [0,1]}$ if the following equivalent conditions hold:

- Let $X_0 \sim \mathbf{p}_0$, and $dX_t = v_t(X_t) \, dt$. Then $\text{Law}(X_t) = \mathbf{p}_t$, or equivalently $X_t \sim \mathbf{p}_t$.
- $(\mathbf{p}_t, v_t)$ satisfies the following **continuity equation**:

$$\partial_t \mathbf{p}_t(x) + \nabla \cdot \big( v_t(x) \, \mathbf{p}_t(x) \big) = 0. \tag{2}$$

In the default setting, we assume that $v_t$ and $\psi_t$ satisfy sufficient regularity conditions so that the above system admits a unique solution. Further details are provided in the appendix.

Note that in the ODE (and SDE) flow generation setting, the flow $\psi$ can be equivalently described by an **interpolation function** $I_t : \mathbb{R}^d \times \mathbb{R}^d \to \mathbb{R}^d$, satisfying

$$I_0(x_0, x_1) = x_0, \quad I_1(x_0, x_1) = x_1. \tag{3}$$

We can then define the probability path (conditional on $X_0, X_1$) as

$$X_t = I_t(X_0, X_1), \quad X_t \sim \mathbf{p}_t.$$

By default, we choose the affine interpolation (Liu et al., 2022; Lipman et al., 2023; 2024): $I_t(x_0, x_1) = (1 - t)x_0 + tx_1$.

### 2.1 CLASSICAL FLOW MATCHING AND CONDITIONAL FLOW MATCHING

The goal of flow matching is to find a neural velocity field $v$ such that $v$ generates the probability path $(\mathbf{p}_t)_{t \in [0,1]}$ with endpoints $\mathbf{p}_0 = \mathbf{p}$ and $\mathbf{p}_1 = \mathbf{q}$.

**Unconditional Flow Matching.** Let $v_t^\theta : [0,1] \times \mathbb{R}^d \to \mathbb{R}^d$ denote a parametrized function (e.g., a neural network). The flow matching loss is defined as

$$\min_{\theta \in \Theta} \mathcal{L}_{\text{FM}}(\theta) = \mathbb{E}_{t, X_t} \big[ \|v_t^\theta(X_t) - v_t(X_t)\|^2 \big], \qquad t \sim \mathcal{U}([0,1]), \quad X_t \sim \mathbf{p}_t, \quad t \perp\!\!\!\perp X_t, \quad (4)$$

where the independence condition $t \perp\!\!\!\perp X_t$ indicates that one first chooses $t \sim \mathcal{U}([0,1])$, and then independently samples $X_t \sim \mathbf{p}_t$.

**Remark 2.1** *In most flow-matching or diffusion-model references, $t$ may be treated either as a random variable (independent of $(X_t)_{t \in [0,1]}$) or as a fixed constant in $[0,1]$. We do not distinguish these cases for convenience.*

In practice, the problem (4) is intractable since the law $\mathbf{p}_t$ is unknown. To address this, the **conditional flow matching**, also known as the **rectified flow** (Liu et al., 2022) objective is used:

$$\mathbb{E}_{(X_0, X_1) \sim \boldsymbol{\pi}_{0,1}} \Big[ \|v_t^\theta(X_t \mid X_0, X_1) - v_t(X_t \mid X_0, X_1)\|^2 \Big], \qquad (5)$$

where the target velocity is given by

$$v_t(X_t \mid X_0, X_1) = \frac{d}{dt} I_t(X_0, X_1) = X_1 - X_0,$$

when the interpolation is affine, i.e. $X_t := I_t(X_0, X_1) = (1-t)X_0 + tX_1$.

## 2.2 Optimal Transport and Related Flow Matching Models

**Optimal Transport.** Let $\mathcal{P}_2(\mathbb{R}^d) := \Big\{ \mathbf{p} \in \mathcal{P}(\mathbb{R}^d) : \int_{\mathbb{R}^d} \|x\|^2 \, d\mathbf{p}(x) < \infty \Big\}$. Given a measurable mapping $T : \mathbb{R}^d \to \mathbb{R}^d$, the pushforward measure $T_\# \mathbf{p}$ is defined as

$$T_\# \mathbf{p}(B) := \mathbf{p}(T^{-1}(B)), \quad \forall B \subseteq \mathbb{R}^d \text{ Borel}, \qquad (6)$$

where $T^{-1}(B) := \{x : T(x) \in B\}$ is the preimage of $B$ under $T$.

Given $\mathbf{p}, \mathbf{q} \in \mathcal{P}_2(\mathbb{R}^d)$, the **optimal transport problem** is

$$OT(\mathbf{p}, \mathbf{q}) := \min_{\boldsymbol{\gamma} \in \Gamma(\mathbf{p}, \mathbf{q})} \int_{\mathbb{R}^d \times \mathbb{R}^d} \|x - y\|^2 \, d\gamma(x, y), \qquad (7)$$

where $\Gamma(\mathbf{p}, \mathbf{q}) := \Big\{ \boldsymbol{\gamma} \in \mathcal{P}(\mathbb{R}^d \times \mathbb{R}^d) : (\pi_1)_\# \boldsymbol{\gamma} = \mathbf{p}, \ (\pi_2)_\# \boldsymbol{\gamma} = \mathbf{q} \Big\}$, with $\pi_1, \pi_2$ denoting the canonical projections. Classical OT theory (Villani, 2003; Villani et al., 2008) guarantees the existence of a minimizer to (7). When the optimal plan $\boldsymbol{\gamma}$ is induced by a mapping $T : \mathbb{R}^d \to \mathbb{R}^d$, that is, $\boldsymbol{\gamma} = (\text{id} \times T)_\# \mathbf{p}$ where $T_\# \mathbf{p} = \mathbf{q}$, the solution is said to be of *Monge form*.

### 2.2.1 Mini-batch Optimal Transport Flow Matching.

The *dynamic* OT, known as the Benamou–Brenier formulation (Benamou & Brenier, 2000) is:

$$OT(\mathbf{p}, \mathbf{q}) = \min_{\{p_t, v_t\}} \int_0^1 \int_{\mathbb{R}^d} \|v_t(x)\|^2 \, d\mathbf{p}_t(x) dt, \qquad \text{(Benamou–Brenier)}$$

$$\text{s.t. } \partial_t \mathbf{p}_t(x) + \nabla \cdot \big(v_t(x), \mathbf{p}_t(x)\big) = 0, \quad \mathbf{p}_0 = \mathbf{p}, \mathbf{p}_1 = \mathbf{q}.$$

Intuitively, dynamic OT finds the most **cost-efficient** probability path with respect to the $\ell_2$ cost.

Inspired by this property, Pooladian et al. (2023); Tong et al. (2023a) adapt OT as the coupling between $\mathbf{p}_0$ and $\mathbf{p}_1$ in (5). The resulting method is called **mini-batch optimal transport flow matching (OT-CFM)**:

$$\mathcal{L}_{\text{OT-CFM}}(\theta) := \mathbb{E}_{\substack{X_0^B \overset{\text{i.i.d.}}{\sim} \mathbf{p}, \\ X_1^B \overset{\text{i.i.d.}}{\sim} \mathbf{q}}} \mathbb{E}_{(X_0, X_1) \sim \boldsymbol{\pi}_{0,1}} \big[ \|v_t^\theta(X_t) - v_t(X_t \mid X_0, X_1)\|^2 \big], \qquad (8)$$

where $B \in \mathbb{N}$, and $\boldsymbol{\pi}_{0,1}$ is the optimal coupling in $OT(\mathbf{p}^B, \mathbf{q}^B)$ with empirical laws

$$\mathbf{p}^B = \text{Law}(X_0^B), \qquad \mathbf{q}^B = \text{Law}(X_1^B). \qquad (9)$$

The term **mini-batch** refers to the fact that the OT coupling $\boldsymbol{\pi}_{0,1}$ is computed from sampled mini-batches $X_0^B$ and $X_1^B$. Compared to using the full coupling $OT(\mathbf{p}, \mathbf{q})$, the mini-batch approach improves training efficiency and introduces stochasticity into the model.

### 2.2.2 Other OT-based Flow Matching Models

Beyond the models described above, several works have extended flow matching by incorporating alternative OT formulations. For example, Klein et al. (2023) combine Gromov–Wasserstein (GW) distance with rectified flow matching, enabling the model to align distributions with heterogeneous supports (e.g., graphs versus point clouds). This direction leverages the structural matching ability of GW to define flow trajectories in non-Euclidean domains.

More recently, Chapel et al. (2025) proposed differentiable generalized sliced OT (GSOT) plans and integrated them with flow matching. By learning nonlinear projections that define generalized sliced Wasserstein distances, their approach inherits both computational scalability and expressive power, allowing efficient flow training on high-dimensional data. Similarly, Tran et al. (2025) applied tree-sliced Wasserstein distances with nonlinear projections to diffusion models, showing that projection-based OT relaxations can improve sampling quality.

Another line of research focuses on using dual formulation or regularized OT formulations. Tong et al. (2023b) combined stochastic interpolations with OT couplings, including entropic OT, leading to the Schrödinger Bridge Flow Matching model. Kornilov et al. (2024) proposed Optimal Flow Matching, which uses the dual formulation of quadratic OT and constrains velocity fields to gradients of convex potentials.

In addition, the Wasserstein Flow Matching framework (Haviv et al., 2024) employs OT and Bures–Wasserstein distances to define pairwise displacements between probability measures (e.g., between shapes), broadening the applicability of flow matching beyond Euclidean metrics.

Overall, these works illustrate that OT can enrich flow matching models in diverse ways: by incorporating structural similarity (GW), scalable projections (sliced OT), dynamic formulations (SB), or convex dual structures (OFM).

### 2.3 Mean-Flow Model

The inference (data generation) step of classical FM requires solving an integration of the form

$$x_1 = x_0 + \int_0^1 v(t, x_t)\, dt, \tag{10}$$

which typically necessitates multiple numerical steps. In Geng et al. (2025), the authors propose the *Mean-Flow model*, which directly learns the *average* vector field:

$$u(t, r, x_t) := u_{t,r}(x_t) := \frac{1}{t-r} \int_r^t v(\tau, x_\tau)\, d\tau, \quad r < t. \tag{11}$$

It is straightforward to verify that $u_{t,r}$ satisfies the following PDE (when $t, r$ are independent):

$$u(t, r, x_t) = v(x_t, t) - (t - r)\Big(v(x_t, t)\, \partial_{x_t} u(t, r, x_t) + \partial_t u^\theta(t, r, x_t)\Big). \tag{12}$$

This leads to the training loss:

$$\mathcal{L}_{\mathrm{MF}}(\theta) := \mathbb{E}_{(X_0, X_1)} \left[\|u_t^\theta(x_t, r, t) - \mathrm{sg}(u_{\mathrm{tgt}}(v_t, t, r))\|^2\right], \tag{13}$$

$$u_{\mathrm{tgt}}(v_t, t, r) = v(x_t, t) - (t - r)\left(v(x_t, t)\, \partial_{x_t} u^\theta(t, r, x_t) + \partial_t u^\theta(t, r, x_t)\right), \tag{14}$$

where sg denotes the stop-gradient operator (i.e., no gradients propagate through this argument with respect to $\theta$). Intuitively, one can view $u_t^\theta$ as the model at the previous moment; thus, $u^\theta$ is not included as input to $u_{\mathrm{tgt}}$. At inference time, the learned mean flow can be directly applied to a base sample $x_0 \sim \mathbf{p}$ in a single step:

$$x_1 \approx x_0 + u_{1,0}(x_0), \tag{15}$$

thereby bypassing multi-step ODE integration. This one-step transport significantly accelerates sampling while maintaining competitive generation quality, showing that generative flows can be effectively compressed into a single mean displacement (Geng et al., 2025).

---

**Algorithm 1** Mean-Flow Training with OT

---
**Input:** Source data $\mathcal{D}_0$ (default to $\mathcal{N}(0, I_d)$), target data $\mathcal{D}_1$, epochs $E$, batch size $B$
**Output:** Trained parameters $\theta$
1: Initialize $\theta$;
2: **for** $e = 1 \rightarrow E$ **do**
3:      **for** mini-batches $(X_0^B, X_1^B) \sim (\mathcal{D}_0, \mathcal{D}_1)$ of size $B$ **do**
4:          Solve OT plan $\gamma = OT(\text{Law}(X_0^B), \text{Law}(X_1^B))$ or OT variants (see e.g. (17), or (20), or (53))
5:          Sample $(X_0, X_1) \sim \gamma$, size$(X_0, X_1) \leq B$ (in default $= B$).
6:          Sample $t, r \sim \mathcal{U}(0, 1)$ with $r \leq t$.
7:          Compute $X_t \leftarrow I_t(X_0, X_1), v_t \leftarrow \frac{d}{dt}(X_0, X_1)$. In default, $X_t = (1-t)X_0 + tX_1, v_t = X_1 - X_0$
8:          Compute $u_{tgt}(v_t, t, r)$ from (14)
9:          $\mathcal{L}(\theta) = \|u^\theta(t, r, x_t) - \text{sg}(u_{tgt})\|^2$
10:         Update $\theta$ based on $\mathcal{L}(\theta)$, e.g. gradient descent, momentum method, etc.
11:      Stop if converges

---

# 3 OUR METHOD: OT-MEAN FLOW

Our OT-mean flow matching is defined as follows:

$$\mathcal{L}_{\text{OTMF}}(u^\theta) := \mathbb{E}_{X_0^B \sim \mathbf{p}, X_1^B \sim \mathbf{q}} \mathbb{E}_{(X_0, X_1) \sim \boldsymbol{\pi}_{0,1}, t} \left[ \|u_t^\theta(t, r, X_t) - u_{\text{tgt}}(v_t, t, r)\|^2 \right], \tag{16}$$

$$\boldsymbol{\pi}_{0,1} \text{ is an optimal plan for } OT(\mathbf{p}^B, \mathbf{q}^B), \quad \mathbf{p}^B = \text{Law}(X_0^B), \ \mathbf{q}^B = \text{Law}(X_1^B).$$

The inference process is the same as in the classical mean-flow model (15).

The above formulation can be viewed as a unified formulation that combines the mini-batch OT flow and the mean flow method. Our method is summarized in Algorithms 1 and 2. We further accelerate training by applying the following OT techniques to compute the batch coupling $\pi_{0,1}$.

## 3.1 OT ACCELERATION METHODS

The computational cost of solving the discrete OT problem via network flow or linear programming is prohibitively high ($\mathcal{O}(n^3 \log n)$) in the worst case for $n$ samples). In the semi-discrete and continuous settings, the complexity can be even worse. To accelerate computation, several approximate OT variants have been proposed. Below we briefly review some of the most widely used approaches.

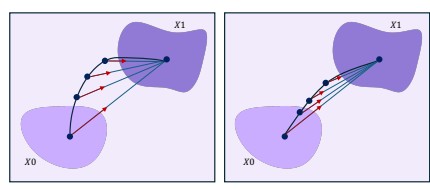

Figure 1: Velocity visualization of a pair of points from the source and target distributions. The straight line denotes the average velocity from an intermediate time to $t = 1$. The OT-MF trajectory is noticeably straighter compared to the vanilla Mean Flow.

**Sinkhorn OT (Entropic Regularization).** A popular relaxation is the entropically regularized OT problem (Cuturi, 2013). For two empirical measures $\mathbf{p} = \sum_{i=1}^n p_i \delta_{x_i}$ and $\mathbf{q} = \sum_{j=1}^m q_j \delta_{y_j}$ with cost matrix $C \in \mathbb{R}^{n \times m}$, the entropic OT problem is

$$\min_{\pi \in \Pi(p,q)} \langle C, \pi \rangle + \varepsilon \, \text{KL}(\pi \,\|\, p \otimes q)$$

$$= \epsilon \text{KL}(\pi \,\|\, e^{-C/\varepsilon} p \otimes q) + \text{constant}, \tag{17}$$

where $\text{KL}(\gamma \,\|\, p \otimes q) := \sum_{i,j} \gamma_{i,j} \ln \frac{\gamma_{i,j}}{p_i q_j}$ is the KL divergence term. The solution is computed efficiently by the Sinkhorn–Knopp algorithm with $\mathcal{O}(n^2)$ cost per iteration.

*Dynamic Schrödinger Bridge View.* Entropic OT also admits a dynamic formulation known as the Schrödinger bridge problem (Léonard, 2014; Chen et al., 2021). It seeks the most likely stochastic process interpolating between $\mathbf{p}$ and $\mathbf{q}$ under a prior Brownian motion. Formally, it solves

$$\min_{P \in \mathcal{P}([0,1] \times \mathbb{R}^d)} \text{KL}(P \,\|\, W_\epsilon) \quad \text{s.t. } P_0 = \mathbf{p}, \ P_1 = \mathbf{q}, \tag{18}$$

where $W_\epsilon$ is the law of the **Wiener process**, $dX_t = \sqrt{\epsilon} dB_t$. As $\varepsilon \rightarrow 0$, the Schrödinger bridge converges to the classical Benamou–Brenier dynamic OT (Benamou–Brenier).

**Linear Optimal Transport.**   Another line of work considers *linearized* OT (Wang et al., 2013) variants, which approximate the quadratic-cost OT by projecting the measures into a linear (Hilbert) subspace of the $L^2$ function space: $\mathcal{L}_2(\mathbb{R}^d; \mathbb{R}^d) := \left\{ f : \mathbb{R}^d \to \mathbb{R}^d : \int_{\mathbb{R}^d} \|f(x)\|^2 d\mu(x) < \infty \right\}$.

In particular, these methods fix a reference measure $\boldsymbol{\sigma} \in \mathcal{P}_2(\mathbb{R}^d)$ (also referred to as the *pivot* measure) and define $\boldsymbol{\gamma}^1, \boldsymbol{\gamma}^2$ as the optimal transportation plans for $OT(\boldsymbol{\sigma}, \mathbf{p})$ and $OT(\boldsymbol{\sigma}, \mathbf{q})$, respectively. The Linear OT plan between $\mathbf{p}$ and $\mathbf{q}$ is then constructed from these *conditional* plans.[1]

$$\boldsymbol{\gamma}_{\mathrm{LOT-lr}} := (\boldsymbol{\gamma}^1_{\cdot 1 | s} \otimes \boldsymbol{\gamma}^2_{\cdot 2 | s})_{\#} \boldsymbol{\sigma}, \qquad \boldsymbol{\gamma}_{\mathrm{LOT-hr}} := \gamma^*(\boldsymbol{\gamma}^1_{\cdot 1 | s}, \boldsymbol{\gamma}^2_{\cdot 2 | s})_{\#} \boldsymbol{\sigma}, \tag{19}$$

where $\boldsymbol{\gamma}^1_{\cdot | s}$ denotes the conditional probability measure of $\boldsymbol{\gamma}^1$ given the first component is $s \in \mathbb{R}^d$, and $\gamma^*(\boldsymbol{\gamma}^1_{\cdot 1 | s}, \boldsymbol{\gamma}^2_{\cdot 2 | s})$ is the optimal coupling for $OT(\boldsymbol{\gamma}^1_{\cdot 1 | s}, \boldsymbol{\gamma}^2_{\cdot 2 | s})$.

It is straightforward to verify that $\boldsymbol{\gamma}_{\mathrm{LOT-lr}}, \boldsymbol{\gamma}_{\mathrm{LOT-hr}}$ are couplings between $\mathbf{p}$ and $\mathbf{q}$. The plan $\boldsymbol{\gamma}_{\mathrm{LOT-lr}}$ is related to the **Low-Rank OT plan** (Scetbon & Cuturi, 2022; Scetbon et al., 2022); similarly, $\boldsymbol{\gamma}_{\mathrm{LOT-hr}}$ is a special case of the Hierarchical OT plan (Halmos et al., 2025).

In the discrete case, suppose $\boldsymbol{\sigma} = \sum_{i=1}^r \sigma_i \delta_{s_i}$, $\mathbf{p} = \sum_{i=1}^n p_i \delta_{x_i}$, and $\mathbf{q} = \sum_{i=1}^m q_i \delta_{y_i}$. Then $\boldsymbol{\gamma}^1 \in \mathbb{R}_+^{r \times n}$ and $\boldsymbol{\gamma}^2 \in \mathbb{R}_+^{r \times m}$, and the above plan reduces to

$$\begin{cases} \boldsymbol{\gamma}_{\mathrm{LOT-lr}} = (\boldsymbol{\gamma}^1)^\top \operatorname{diag}(1/\sigma) \, \boldsymbol{\gamma}^2, \\ [\boldsymbol{\gamma}_{\mathrm{LOT-hr}}]_{\mathcal{D}(\boldsymbol{\gamma}^1[i,:]) \times \mathcal{D}(\boldsymbol{\gamma}^2[i,:])} = \sigma_i \gamma^*(\boldsymbol{\gamma}^1_{\cdot | s_i}, \boldsymbol{\gamma}^2_{\cdot | s_i}), \quad \forall i \in [1:r], \end{cases} \tag{20}$$

where in the second plan, $\mathcal{D}(v) := \{i : v_i > 0\}$.

The computational complexity of the low-rank linear OT coupling is $\mathcal{O}(rn(r+n))$, while that of the hierarchical linear OT coupling is $\mathcal{O}(rn(r+n) + r(n/r)^3)$. When $r$ is small, the low-rank formulation yields a significant reduction in complexity. When both $r$ and $n/r$ are small (i.e., when $n$ admits a suitable factorization), the hierarchical method also achieves reduced complexity.

## 4 EXPERIMENTS

We evaluate the empirical benefits of Transport-based Flows on four generative modeling tasks: (a) controlled low-dimensional synthetic data, (b) image generation, (c) image-to-image translation, and (d) point cloud generation.   In addition, we test several other optimal transport variants within the Mean Flow framework. Some of these introduce uncertainty into the transport problem (e.g., Sinkhorn), while others focus on improving

---

**Algorithm 2** Inference:  Flow-Matching ODE Integration

---

**Input:** Trained mean vector field $u_\theta(x, t, r)$; steps $T$; size $n$
**Output:** Sample $x_1$
 1: Sample $n$ i.i.d. $x_0 \sim \mathcal{D}_0$, set $x_t = x_0$
 2: **for** $t = 1/T, 2/T, \ldots, 1$ **do**
 3:     $s = t - 1/T, \quad x_t \leftarrow x_t + u^\theta(x_t, t, s)$
 4: $x_1 \leftarrow x_t$

---

computational efficiency (e.g., LOT-LR, LOT-HR). We further demonstrate that one-step generation can be enhanced by incorporating optimal transport–based sampling strategies. Full implementation details are provided in Appendix C.

**OT solver setup.** For vanilla OT and low-rank OT, we use the C++ linear programming solver provided in the PythonOT library (Flamary et al., 2021).  For Sinkhorn, we evaluate three implementations: (i) the Python implementation in PythonOT (supports both CPU and GPU), (ii) a Numba-accelerated CPU version,[2] and (iii) the JAX-based implementation in the OTT-JAX library (Cuturi et al., 2022). For each experiment, we report results using the fastest implementation.

### 4.1 TOY EXAMPLE: CONTROLLED LOW-DIMENSIONAL POINT CLOUDS

**Dataset.** We first present results on synthetic toy examples, considering five distribution pairs: a Gaussian $(N) \to$ a mixture of 8 Gaussian (8-Gaussians); the half-moons dataset (Zhou et al., 2004)

---

[1]The original Linear OT plan is formally defined through an optimization problem; the low-rank construction presented here is a practically convenient alternative. Under suitable regularity conditions, the two formulations coincide. We refer the reader to Moosmüller & Cloninger (2020), Bai et al. (2023), and Rabbi et al. (2024) for details.

[2]https://numba.pydata.org/

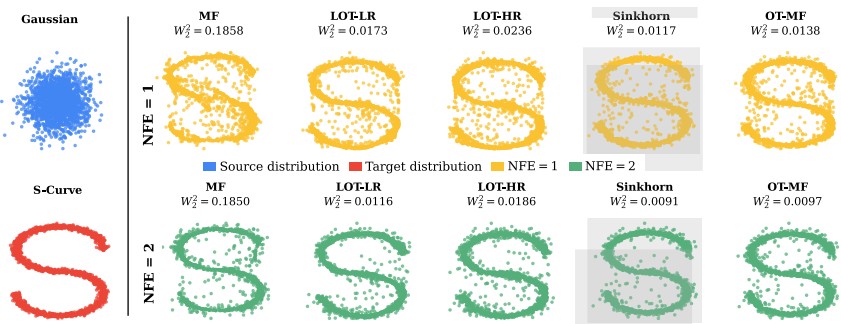

Figure 2: Comparison of different transport-based mean flows for $N \to S-$curve. The first row is NFE=1, the second row is NFE=2. We report the 2-Wasserstein distance between the predicted distribution and the target distribution.

(denoted as *moons*) $\to$ 8-Gaussians; $N \to$ moons; $N \to$ the S-curve dataset (Pedregosa et al., 2011) (denoted as *scurve*); and $N \to$ the checkerboard dataset (Dinh et al., 2017).

**Models and training setup.** Following Geng et al. (2025), we use a 3-layer MLP as the generator and utilize *Adam* optimizer with learning rate lr $= 1e - 3$. The results for two of these experiments are presented in Figure 2.

### Evaluation and Performance

Since the source and target data are 2D point clouds, we present the Wasserstein 2 distance (Villani, 2003) as a metric. From Table 1 and Figure 2 we observe that OT-based mean flow methods significantly improve upon the vanilla mean flow. Other OT variants, such as Sinkhorn and LOT, also demonstrate improved performance. In particular, LOT enhances computational efficiency compared to the original OT while maintaining relatively high accuracy.

Table 1: Comparison of different transport-based mean flows over five distribution pairs at NFE=1/2 (denoted as @1 and @2) average over three random seeds. Best per column is bold gray.

| Dataset $\to$ | $N\to8gaussians$ | | | $moons\to8gaussians$ | | | $N\to moons$ | | | $N\to scurve$ | | | $N\to checkerboard$ | | |
|---|---|---|---|---|---|---|---|---|---|---|---|---|---|---|---|
| Method $\downarrow$ Metric $\to$ | $W_2^2$@1 | $W_2^2$@2 | TR(ms) | $W_2^2$@1 | $W_2^2$@2 | TR(ms) | $W_2^2$@1 | $W_2^2$@2 | TR(ms) | $W_2^2$@1 | $W_2^2$@2 | TR(ms) | $W_2^2$@1 | $W_2^2$@2 | TR(ms) |
| MF | 0.3931 | 0.3121 | **4.91** | 0.5601 | 0.5435 | **4.82** | 0.0719 | 0.0891 | **4.93** | 0.1913 | 0.1855 | **4.83** | 0.0721 | 0.0654 | **4.80** |
| LOT-LR | 0.0683 | 0.0539 | 8.82 | 0.0801 | 0.0657 | 8.27 | 0.0320 | 0.0250 | 8.47 | 0.0164 | 0.0117 | 8.30 | 0.0179 | 0.0168 | 8.38 |
| LOT-HR | 0.0268 | 0.0214 | 10.18 | 0.0648 | 0.0559 | 10.24 | 0.0322 | 0.0272 | 10.22 | 0.0140 | 0.00958 | 10.17 | 0.00733 | 0.00648 | 10.12 |
| Sinkhorn | 0.0145 | 0.0107 | 21.1 | **0.0148** | **0.0113** | 21.3 | **0.0212** | **0.0149** | 21.6 | **0.00747** | **0.00432** | 21.2 | **0.00473** | **0.00411** | 21.1 |
| OT-MF | **0.0141** | **0.0104** | 12.4 | 0.0166 | 0.0120 | 12.6 | 0.0241 | 0.0165 | 13.0 | 0.00842 | 0.00472 | 12.2 | 0.00510 | 0.00456 | 12.1 |

## 4.2 IMAGE GENERATION

**Dataset Setup.** We study one-step MeanFlow generation on MNIST in the latent space of a pretrained VAE tokenizer Rombach et al. (2022). Each $28 \times 28$ digit is padded to $32 \times 32$, normalized to $[-1, 1]$, replicated across three channels, and encoded once by the frozen VAE into $4 \times 4 \times 4$ latents, which are cached for training.

Table 2: Comparison of Transport-based Flows for image generation on MNIST across NFEs. We report FID and $W_2$ for 1/2/5/10 steps (EMA=True). Best per column is bold with gray background.

| Metric $\to$ | FID $\downarrow$ | | | | $W_2 \downarrow$ | | | |
|---|---|---|---|---|---|---|---|---|
| Method $\downarrow$ / NFE $\to$ | 1 | 2 | 5 | 10 | 1 | 2 | 5 | 10 |
| MF | 3.6709 | 1.0880 | 0.6318 | 0.7267 | 8.7449 | 8.2560 | 8.0634 | 8.0602 |
| LOT-LR | 2.2258 | 0.7449 | 0.5371 | 0.5531 | 8.4315 | 8.1481 | 8.0312 | 7.9941 |
| LOT-HR | 1.9754 | 0.8357 | 0.6053 | 0.5759 | 8.3815 | 8.1922 | 8.0683 | 8.0138 |
| Sinkhorn | 3.6944 | 1.0782 | 0.6362 | 0.7135 | 8.6554 | 8.2903 | 8.1000 | 8.1040 |
| OT-MF | **1.9179** | **0.6123** | **0.4689** | **0.4935** | **8.2102** | **8.0383** | **8.0029** | **7.9546** |

**Network Model and Settings.** Our generator uses a ConvNeXt-style U-Net (Geng et al., 2025), adapted to the low-resolution latent tensor ($\sim$59M parameters), with dual sinusoidal embeddings for flow time $t$ and solver step size $h$. Training largely follows Geng et al. (2025), with *Adam* ($10^{-3}$ lr, $(0.9, 0.99)$, batch 256, no weight decay), 30k iterations, 10% warm-up, EMA (0.99, every 16 steps), and logit-normal timestep sampling ($P_{mean}^t = -0.6$, $P_{std}^t = 1.6$, $P_{mean}^r = -4.0$, mismatch 0.75). We use the JVP-based loss with adaptive reweighting, and evaluate independent pairing as well as transport-based pairings (OT, Sinkhorn OT, LOT-LR, LOT-HR).

**Evaluation Metric.** We evaluate OT solvers for Mean Flows with Euler integration across 1–10 NFEs. Performance is measured using Fréchet Inception Distance (FID) (Heusel et al., 2017) and 2-

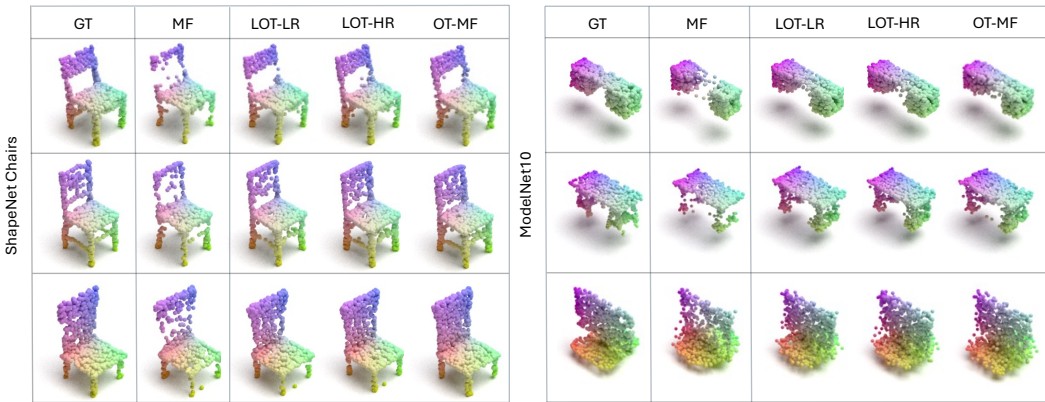

Figure 3: Single step (NFE=1) sample generation on *ShapeNet* Chairs and *ModelNet10*

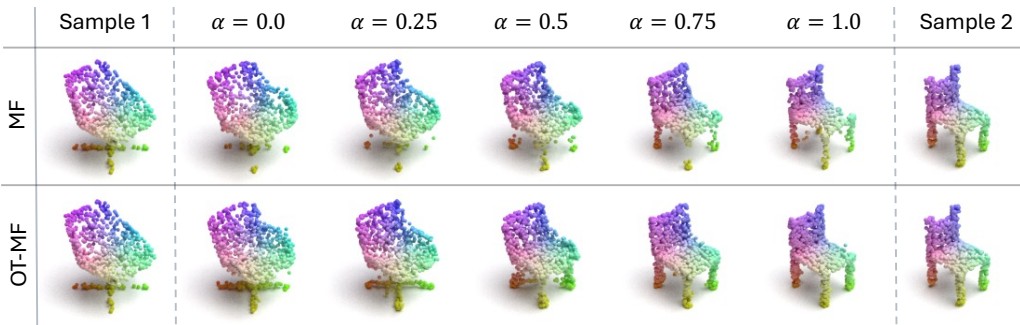

Figure 4: Single step (NEF=1) shape interpolation on two samples from *ShapeNet* chairs. $\alpha \in [0, 1]$ controls the interpolation of the source and target features.

Wasserstein distance ($W_2$) (Villani et al., 2008) between generated and VAE-reconstructed images, computed in pixel space with the Inception network from `TorchMetrics`.

**Consolidated generation results (NFE=1/2/5/10).** Table 2 shows that the exact OT solver achieves the best FID and $W_2$ across NFEs; LOT-HR and LOT-LR are competitive at NFE=1. Performance improves markedly from NFE 1→5 and plateaus by NFE=10.

### 4.3 POINT CLOUD GENERATION AND INTERPOLATION

**Experimental setup.** We train and evaluate point cloud generation on a subset of *ShapeNet* (Chang et al., 2015), a large-scale dataset of 3D CAD models, and *ModelNet10* (Wu et al., 2015), which contains 10 object classes. For our experiments, we use the *Chair* class from *ShapeNet*. Following standard practice, each object is preprocessed by uniformly sampling point clouds from mesh surfaces. We utilize a pre-trained PointNet-based (Qi et al., 2017) autoencoder to extract a vector representation of each point cloud. This is then used to condition our flow model during generation. We provide additional details of our experimental setup in section D.2.

**Results.** Figure 3 shows one-step generation (NFE=1) results for MeanFlow, OT-MeanFlow, LOT-HR, LOT-LR, and the ground truth on *ShapeNet* Chairs and *ModelNet10* (classes 'desk', 'table', and 'monitor'). Incorporating OT-based sampling enables the models to capture finer details and generate more accurate shapes. Table 3 reports the average Wasserstein-2 distance and training time per epoch. All OT-augmented variants outperform Mean-Flow, with OT-MF achieving the best performance while introducing only moderate additional cost.

Table 3: $W_2$ and Average Train Time per epoch (TR) reported on *ShapeNet* (SN) Chairs and *ModelNet10*. Best values are bold and gray, followed by second-bests denoted by underline.

| Method | SN Chairs | | ModelNet10 | |
|---|---|---|---|---|
| | $W_2$ | TR(s) | $W_2$ | TR(s) |
| MF | 0.0477 | **16.32** | 0.0377 | **23.41** |
| LOT-LR | 0.0168 | 17.22 | 0.0231 | 24.82 |
| LOT-HR | 0.0141 | 18.24 | 0.0227 | 26.52 |
| OT-MF | **0.0121** | 20.81 | **0.0208** | 28.64 |

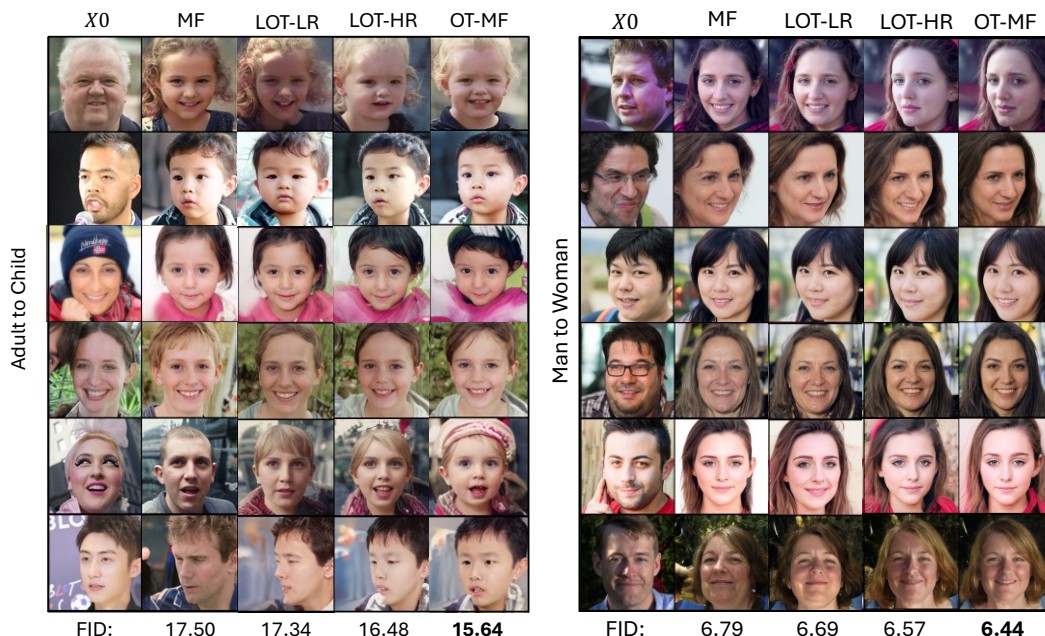

Figure 5: Comparison of one-step mean flow method for Image-to-Image translation on *Adult→Child* and *Man→Woman*.

**Shape Interpolation.** To further analyze the effect of our proposed method, we report shape interpolation results in Figure 4. We randomly sample two shapes from the *ShapeNet* Chairs test data, and use a convex combination of the context features of the two shapes to generate new interpolated shapes in a single step. As shown in the figure, OT-MF can capture details more precisely in an interpolated setting as well, resulting in higher quality shapes. In particular, we observe that OT-MF induces a smoother interpolation, while vanilla MF exhibits relatively poor performance—for example, the leg of the interpolated chair appears distorted.

### 4.4 Unpaired Image-to-Image Translation

Next, we evaluate our method on unpaired image-to-image translation (Zhu et al., 2017), using dataset splits from Korotin et al. (2023) and (Gushchin et al., 2024). The dataset is derived from FFHQ (Karras et al., 2019), with 60k training and 10k test images. All images are encoded into a 512-dimensional latent space using ALAE (Pidhorskyi et al., 2020), and train flow models to transform the latents corresponding to a set of source images to latents corresponding to a set of target images. We compare OT, LOT-HR, and LOT-LR against vanilla MF on the splits *adult→child* and *man→woman*. Evaluation uses FID Heusel et al. (2017) between reconstructed autoencoder images and model outputs. Figure 5 reports qualitative results and FID scores, showing OT-MF achieves the best performance, followed by LOT variants for one-step generation.

### 5 Summary

We propose a new one-step flow matching framework that unifies optimal transport conditional flow matching and mean flow matching under a common formulation. By leveraging optimal transport couplings, our method provides a principled way to construct target average velocity fields that better capture the geometric structure of the data. We further explore approximate OT variants such as low-rank and hierarchical refinements, which offer improved computational efficiency without sacrificing performance.

Through extensive experiments on point cloud and image generation, as well as image-to-image translation tasks, we demonstrate that OT-based mean flow methods consistently yield more robust and higher-quality results for one-step generative modeling compared to vanilla mean flow. Our study highlights the potential of integrating optimal transport with one-step flow-based generative modeling, offering both theoretical insights and practical improvements.

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

## A  DEFAULT NOTATION AND CONVENTION

### SPACES, MEASURES, VECTORS, FUNCTIONS

- $\mathbb{R}^d$: $d$-dimensional Euclidean space with inner product $\langle \cdot, \cdot \rangle$ and norm $\| \cdot \|$.
- $\mathcal{P}(\mathbb{R}^d)$: Set of Borel probability measures on $\mathbb{R}^d$.
- $\mathcal{P}_2(\mathbb{R}^d)$: Probability measures with finite second moment, i.e. $\{\mu \in \mathcal{P}(\mathbb{R}^d) : \int \|x\|^2 d\mu(x) < \infty\}$.
- $\delta_x$: Dirac measure at $x$.
- $\text{Supp}(\mu)$: Support of a measure $\mu$.
- $T_{\#}\mu$: Pushforward of $\mu$ by $T : \mathbb{R}^d \to \mathbb{R}^d$, defined by $(T_{\#}\mu)(B) = \mu(T^{-1}(B))$.
- $1_n$: $n$-dimensional vector of all ones.
- $M_{A \times B}$: For $M \in \mathbb{R}^{n \times m}$ and $A \subset [1 : n], B \subset [1 : m]$, the submatrix $[M_{i,j}]_{i \in A, j \in B}$.

### Random Variables and Probabilities

- $\mathbf{p} \in \mathcal{P}(\mathbb{R}^d)$: Source distribution (default $\mathbf{p} = \mathcal{N}(0, I_d)$).
- $p$: Probability density or mass function of $\mathbf{p}$. *For convenience, in some parts of the article, we do not distinguish measure $\mathbf{p}$ and its density/mass function $p$.*
- $\mathbf{q} \in \mathcal{P}(\mathbb{R}^d)$: Target (data) distribution; in practice, approximated by the training dataset.
- $X_0 \sim \mathbf{p}$, $X_1 \sim \mathbf{q}$: Source and target random variables (realizations of $\mathbf{p}, \mathbf{q}$).
- $\text{Law}(X)$: Distribution of random variable $X$.
- $\boldsymbol{\pi}_{0,1}, \boldsymbol{\gamma}$: Coupling measures with marginals $\mathbf{p}, \mathbf{q}$.
- $\gamma \in \mathbb{R}^{n \times m}$: Probability mass function of $\boldsymbol{\gamma}$ when $p, q$ are discrete of sizes $n, m$.
- $\boldsymbol{\gamma}_1, \boldsymbol{\gamma}_2$: First and second marginals of $\boldsymbol{\gamma}$.
- $\gamma_1, \gamma_2$: pmfs of $\boldsymbol{\gamma}_1, \boldsymbol{\gamma}_2$, with $\gamma_1 = \gamma 1_n$, $\gamma_2 = \gamma^\top 1_m$.
- $\Gamma(\mathbf{p}, \mathbf{q})$: Set of couplings between $\mathbf{p}$ and $\mathbf{q}$.
- $X_0 \perp\!\!\!\perp X_1$: Independence between $X_0$ and $X_1$.
- $\mathbb{E}[\cdot]$: Expectation (subscripts indicate the distribution if needed).

### ODEs, Flows, Paths, Interpolations

- $(\mathbf{p}_t)_{t \in [0,1]}$: Probability path, i.e. a curve in $\mathcal{P}(\mathbb{R}^d)$.
- $v_t : [0, 1] \times \mathbb{R}^d \to \mathbb{R}^d$: Time-dependent velocity field.
- $\psi_t$: Flow map defined by $d\psi_t(x_0) = v_t(\psi_t(x_0)) \, dt$, with $\psi_0(x_0) = x_0$.
- $X_t = \psi_t(X_0)$: State along the flow; $\text{Law}(X_t) = \mathbf{p}_t$.
- $\partial_t p_t + \nabla \cdot (v_t p_t) = 0$: Continuity equation for $(p_t, v_t)$. (Here we do not distinguish measures from densities/pmfs unless needed.)
- $I_t(x_0, x_1)$: Interpolation between $x_0$ and $x_1$, with $I_0 = x_0, I_1 = x_1$ (default $I_t = (1 - t)x_0 + tx_1$).
- $X_t = I_t(X_0, X_1)$: Interpolation-induced path used in conditional FM.

### Optimal Transport (OT)

- $OT(\mathbf{p}, \mathbf{q})$: Quadratic-cost OT,

$$\min_{\boldsymbol{\gamma} \in \Gamma(\mathbf{p}, \mathbf{q})} \int \|x - y\|^2 \, d\gamma(x, y).$$

- $\boldsymbol{\gamma} = (\text{id} \times T)_{\#}\mathbf{p}$: Monge solution, where $T$ is the transport map with $T_{\#}\mathbf{p} = \mathbf{q}$.

- Benamou–Brenier dynamic formulation:

$$\min_{(\mathbf{p}_t, v_t)} \int_0^1 \int \|v_t(x)\|^2 \, d\mathbf{p}_t(x) \, dt, \quad \mathbf{p}_0 = \mathbf{p}, \ \mathbf{p}_1 = \mathbf{q},$$

  subject to the continuity equation $\partial_t p_t + \nabla \cdot (v_t p_t) = 0$.

- Dual OT formulation: Equivalent characterization in terms of convex potentials.
- Sinkhorn OT: Entropic OT with regularization $\varepsilon > 0$ and cost matrix $C$.
- Sliced OT: OT averaged over 1D projections $\langle \theta, \cdot \rangle$ with $\theta \in \mathbb{S}^{d-1}$.
- Linear OT (LOT): OT linearized via a reference $\boldsymbol{\sigma}$, including low-rank and hierarchical variants.

### Flow Matching (FM) and Mean Flows (MF)

- $t, s \in [0, 1]$: time variable, with $s \leq t$
- $D(\mu, \nu)$: Bregman Divergence with

$$D(x, y) := \Phi(x) - \big[\Phi(v) + \langle x - y, \nabla \Phi(y) \rangle \big].$$

  where $\phi$ is convex function

- $\mathcal{L}_{\text{FM}}$: Unconditional FM loss,

$$\mathbb{E}_{t, X_t} \big[ \|v_t^\theta(X_t) - v_t(X_t)\|^2 \big],$$

  or in general,

$$\mathbb{E}_{t, X_t}[D(v_t^\theta(X_t), v_t(X_t))].$$

- $Z$: auxiliary variables used to construct the conditional velocity field and the conditional flow matching. In this article, we only discuss the cases $Z = X_1$ and $Z = (X_0, X_1)$.
- $\mathbf{p}_Z, \mathbf{p}_{X_0}, \mathbf{p}_{X_1}$: probability measures of $Z, X_0, X_1$. Their probability density/mass function are $p_Z, p_{X_0}, p_{X_1}$.
- $v_t(\cdot | Z)$: The velocity field given variable $Z$.
- $\mathbf{p}_{t|Z}$: the conditional probability path at time $t$ given $Z$.
- $\mathcal{L}_{\text{CFM}}$: Conditional FM loss with $X_t = I_t(X_0, X_1)$ and target $\frac{d}{dt} I_t(X_0, X_1)$. In particular,

$$\mathbb{E}_{t, (X_0, X_1) \sim \boldsymbol{\pi}_{0,1}}[\|v_t^\theta(X_t) - v_t(X|Z)\|^2].$$

  Or in general

$$\mathbb{E}_{t, (X_0, X_1) \sim \boldsymbol{\pi}_{0,1}}[D(v_t^\theta(X_t), v_t(X|Z))].$$

- Mini-batch OT–CFM: Uses $\boldsymbol{\pi}_{0,1}$ from $OT(p^B, q^B)$, where $p^B, q^B$ are empirical batch measures.
- $u_{t,r}$: Mean flow,

$$u_{t,r}(x) = \frac{1}{t - r} \int_r^t v_\tau(x_\tau) \, d\tau, \quad r < t.$$

- $u_{\text{tgt}}$: Mean-flow training target,

$$u_{\text{tgt}} = v - (t - r)\big(v \, \partial_x u^\theta + \partial_t u^\theta\big),$$

  an approximation of the true $u_{t,r}$ (based on sample velocities and the model $u^\theta$ at the "previous moment").

- $x_1 \approx x_0 + u_{1,0}(x_0)$: One-step mean-flow inference.

### Batches and Computational Objects

- $p^B = \frac{1}{B} \sum_{i=1}^B \delta_{x_i}$: Empirical (mini-batch) measure of size $B$.

- $X^B \sim p^B$: A realization of the empirical distribution.

- $C \in \mathbb{R}^{n \times m}$: Pairwise cost matrix, typically $C_{ij} = \|x_i - y_j\|^2$.

- $\mathrm{diag}(\boldsymbol{a}), \boldsymbol{I}$: Diagonal matrix with $\boldsymbol{a}$ on the diagonal; $\boldsymbol{I}$ is the identity matrix.

- $\mathcal{O}(\cdot)$: Asymptotic computational complexity.

### Flow matching under guidance

- $\mathbf{c}$: guidance variable with $\mathbf{c} \sim \mathbf{p_c}$.

- $v(t, \mathbf{x}|\mathbf{c})$: the marginal velocity field conditional on guidance $\mathbf{c}$.

- $v(t, \mathbf{x})$: the marginal velocity field:

$$v(t, \mathbf{x}) = \mathbb{E}_{\mathbf{c}}[v(t, \mathbf{x}|c)] = \mathbb{E}_{\mathbf{c}}\mathbb{E}_{(X_0, X_1) \sim \boldsymbol{\pi}_{0,1}|_{\mathbf{c}}}(X_1 - X_0)$$

- $\omega > 1$: guidance scalar.

- $\eta \in [0, 1]$: parameter controls the weight of (averaged) velocity with and without guidance. In default $\eta = 0$ (means no unconditional velocity).

## B  BACKGROUND:ODE, FLOW MATCHING AND OPTIMAL TRANSPORT

In the main text, we briefly introduced the background of ODEs, flow matching, and mean flows. In this section, we provide a more detailed introduction and a survey: we revisit these concepts in depth and present prior work within a unified, consistent framework to facilitate the reader's understanding.

### B.1  ODE, FLOW AND PROBABLITY PATH.

Given a pair of probability measures $(\mathbf{p}, \mathbf{q})$, where $\mathbf{p}$ is a known source (noise) distribution, $\mathbf{q}$ is an unknown target (data) distribution, and both $\mathbf{p}$ and $\mathbf{q}$ are supported in $\mathbb{R}^d$ for some positive integer $d$.

The goal of **Flow Matching** is to build a **Probability Path** $(\mathbf{p}_t)_{t \in [0,1]}$ such that $\mathbf{p}_0 = \mathbf{p}, \ \mathbf{p}_1 = \mathbf{q}$. In particular, FM aims to train the **Velocity Field** neural network, which generates the probability path $(\mathbf{p}_t)_{t \in [0,1]}$.

We start from the following ODE problem:

$$\begin{cases} \psi : [0, 1] \times \mathbb{R}^d \to \mathbb{R}^d, (t, x_0) \mapsto \psi_t(x_0), \\ v : [0, 1] \times \mathbb{R}^d \to \mathbb{R}^d, (x, t) \mapsto v_t(x), \\ d\psi_t(x_0) = v_t(\psi_t(x_0))dt & \text{(flow ODE)}, \\ \psi_0(x_0) = x_0 & \text{(initial condition)}. \end{cases} \tag{21}$$

Here $v_t$ is called the **time-dependent velocity field**, and the solution $\psi$ is called the **time-dependent flow**.

In the default setting, we suppose $v_t$ satisfies the condition of the following fundamental theorem, which guarantees the existence and uniqueness of $\psi_t$ in (21):

**Theorem B.1** *[Flow existence and uniqueness LaSalle (1968); Perko (2013); Lipman et al. (2024)] If $v : [0, 1] \times \mathbb{R}^d \to \mathbb{R}^d$ is continuously differentiable, then the ODE problem (21) admits a unique solution $\psi$. Furthermore, $\psi_t$ is a diffeomorphism for each $t \in [0, 1]$, i.e. $\psi_t$ is continuously differentiable with a continuously differentiable inverse $\psi_t^{-1}$.*

**Remark B.2** *The above theorem demonstrates that, given a velocity field $v_t$ (with regular conditions), it uniquely determines the flow $\psi_t$. The reverse direction is straightforward: given a continuously differentiable $\psi_t$, we can obtain $v_t$ via $v_t = \frac{d}{dt}\psi_t$. Therefore, velocity fields and flows are equivalent descriptions of the same object.*

We define a set of random variables (vectors):

$$X_t = \psi_t(X_0), \ \mathbf{p}_t = \text{Law}(X_t), \tag{22}$$
$$X_0 = \psi_0(X_0) \sim \mathbf{p}_0 := \mathbf{p}.$$

This means $\mathbf{p}_t$ is the distribution of the random variable $X_t$. The induced probability distribution family $\{\mathbf{p}_t\}_{t \in [0,1]}$ is called the **Probability Path**. Thus, the above ODE reads

$$dX_t = v_t(X_t)\,dt. \tag{23}$$

Another way to describe the relation between $v_t, \mathbf{p}_t$ is the **continuity equation** Villani et al. (2008):

$$\frac{d}{dt}\mathbf{p}_t = \nabla \cdot (v_t\mathbf{p}_t), \quad \forall t \in [0,1]. \tag{24}$$

Note, another equivalent continuity equation is defined by replacing $\mathbf{p}_t$ by its density/pmf $p_t$. For convenience, we do not distinguish them in this article.

**Theorem B.3** *Let $(\mathbf{p}_t)_{t \in [0,1]}$ be a probability path and $v_t$ a locally Lipschitz integrable velocity field. Then the following are equivalent:*

- *$(v_t, \mathbf{p}_t)$ satisfies the continuity equation (24).*

- *$(v_t, X_t)$ satisfies the ODE (23).*

*We say $v_t$ generates the probability path $\mathbf{p}_t$ if one of the above equivalent statements holds, with initial condition $X_0 \sim \mathbf{p}_0$.*

**Remark B.4** *The realizations generated by $v_t$, $\{X_t\}_{t \in [0,1]}$, define a stochastic process, i.e., $(X_t, X_s)$ admits a joint distribution. However, unlike Theorem B.1, given a probability path $\{\mathbf{p}_t\}$, there may exist multiple distinct stochastic processes $\{X_t\}$ such that $\mathbf{p}_t = Law(X_t)$ for all $t$.*

**Flow Matching Problem.** Let $v_t^\theta : [0,1] \times \mathbb{R}^d \to \mathbb{R}^d$ denote a parametrized function (e.g., a neural network). The goal of the flow matching problem, equivalently speaking, the **flow matching loss**, is:

$$\min_{\theta \in \Theta} \mathcal{L}_{\text{FM}}(\theta) = \mathbb{E}_{t,X_t}\big[D(v_t^\theta(X_t), v_t(X_t))\big], \quad t \sim \mathcal{U}([0,1]), \ X_t \sim \mathbf{p}_t, \ t \perp\!\!\!\perp X_t, \tag{25}$$

where $D(\cdot, \cdot)$ is a Bregman divergence. For example, if $\Phi : \mathbb{R}^d \to \mathbb{R}$ is strictly convex, then

$$D(u, v) := \Phi(u) - \big[\Phi(v) + \langle u - v, \nabla\Phi(v)\rangle\big]. \tag{26}$$

## B.2 CONDITIONAL FLOW MATCHING

Following the previous section, we define random variables $(X_0, X_1) \sim \boldsymbol{\pi}_{0,1}$ where $\boldsymbol{\pi}_{0,1}$ is a joint measure with marginals $\mathbf{p}, \mathbf{q}$. For example, $\boldsymbol{\pi}_{0,1}$ can be independent coupling, i.e. $\boldsymbol{\pi}_{0,1} = \mathbf{p} \otimes \mathbf{q}$.

Next, we aim to construct a probability path $(\mathbf{p}_t)_{t \in [0,1]}$ and the related flow model $(v_t, \psi_t)$. Note, this task can be dramatically simplified by adopting a conditional strategy. In particular, we introduce an auxiliary random variable $Z \sim \mathbf{p}_Z$ (in general, $Z$ only depends on $X_0, X_1$, i.e. $Z \in \sigma(X_0, X_1)$ where $\sigma(X_0, X_1)$ is the $\sigma-$field defined by $X_0, X_1$.

For example $Z = X_1$ or $Z = (X_0, X_1)$).

### B.2.1 CONDITIONAL FLOW MATCHING IN THE GENERAL CASE

Given an auxiliary random variable $Z \sim \mathbf{p}_Z$, we consider the conditional path $\mathbf{p}_{t|Z}(\cdot|z)$, and the induced marginals

$$p_t(x) = \int p_{t|Z}(x|z)p_Z(z)dz. \tag{27}$$

Similarly, suppose $v_{t|Z}(\cdot|z)$ generate $p_{t|Z}(\cdot|z), \forall z$. Similar to marginal probability distribution, we set the **marginal velocity field**:

$$v_t = \mathbb{E}[v_t(X_t|Z)|X_t = x],. \tag{28}$$

**Theorem B.5** *[Marginal and Conditional velocity fields Lipman et al. (2024)] Suppose* $(\mathbf{p}_{t|Z}(\cdot|z), v_t(\cdot|z))$ *satisfies some regular conditions, that is,* $C_1([0,1)\mathbb{R}^d)$ *and* $v_t(x|z)$ *is* $C_1([0,1) \times \mathbb{R}^d, \mathbb{R}^d)$ *as a function of* $(t,x)$. *Furthermore,* $\mathbf{p}_Z$ *has compact support. Finally,* $\mathbf{p}_t(x) > 0$ *for all* $x \in \mathbb{R}^d$ *and* $t \in [0,1)$.

*Thus, if* $v_{t|Z}(\cdot|z)$ *is integrable and it generates* $p_{t|Z}(\cdot|z)$ *for each* $z$, *then* $v_t$ *defined in (28) generates* $p_t$ *defined in (27).*

Based on it, we can propose the **conditional flow matching** model:

$$\mathcal{L}_{CFM}(\theta) := \mathbb{E}_{t, Z \sim P_Z, X_t \sim p_{\cdot|Z}} D(v_t(X_t|Z), u_\theta^t(X_t)). \tag{29}$$

And the following theorems can demonstrate the equivalence between the Flow matching and conditional flow matching problems (25) and (29):

**Theorem B.6** *Under the conditions of B.5 we have the following:*

$$\nabla_\theta \mathcal{L}_{FM}(\theta) = \nabla_\theta \mathcal{L}_{CFM}(\theta) \tag{30}$$

**Proposition B.7 (Liu et al. (2022))** *Under the conditions of B.5, the population solution of the conditional flow matching problem is given by (28).*

Furthermore, the dynamic generated by $v_t$ (28) is called **rectified flow** in Liu et al. (2022).

### B.2.2  CONDITIONAL FLOW ON $X_1$

In this section, we set:

$$Z = X_1.$$

We consider a mapping

$$[0,1] \times \mathbb{R}^d \ni (t,x) \mapsto \psi_t(x|x_1) \in \mathbb{R}^d$$

that satisfies the following conditions: for each $x_1$, we have

$$\begin{cases} \psi_0(x|x_1) = x, \\ \psi_1(x|x_1) = x_1, \\ \psi_t(\cdot|x_1) \text{ is a diffeomorphism.} \end{cases} \tag{31}$$

By setting the random variables $X_t|_{X_1 = x_1} = \psi_t(X_0|x_1)$, we obtain

$$\text{Law}(X_t|_{X_1 = x_1}) = p_{t|1}(\cdot|x_1) := \psi_t(\cdot|x_1)_\# \pi_{0|1}(\cdot|x_1),$$

which defines a conditional probability path. One can verify that the following boundary conditions are satisfied:

$$p_{0|1}(\cdot|x_1) = \pi_{0,1}(\cdot|x_1), \quad p_{1,1}(\cdot|x_1) = \delta(\cdot, x_1). \tag{32}$$

By Theorem B.1, the following mapping

$$v_t(x|x_1) := \dot{\psi}_t(x_0|x_1) = \dot{\psi}_t(\psi^{-1}(x|x_1)|x_1), \quad \forall x \text{ such that } x = \phi_t(x_0) \text{ for some } x_0 \in \text{Supp}(X_0),$$

is the unique velocity field that generates the conditional path $(p_t(\cdot|x_1)), \forall x_1$.

**Remark B.8** *In some literature (e.g., Haxholli et al. (2024)),* $p_{t|1}(\cdot)$ *or* $v_t(\cdot|1)$ *are introduced first, and the boundary conditions for the (conditional) flow mapping* $\psi_t(\cdot|x_1)$ *are then derived. Intuitively, describing the conditional flow via* $\phi_t(\cdot|x_1)$, $v_t(\cdot|x_1)$, *or* $p_{t|1}(\cdot|x_1)$ *is equivalent, as established by the fundamental theorem B.1. Here, we follow the convention introduced in Lipman et al. (2024).*

Based on the above setting, the conditional flow training loss (29) becomes:

$$\begin{aligned} \mathcal{L}_{CFM}(\theta) &:= \mathbb{E}_{t, X_1, X_t \sim p_{\cdot|X_1}} D(v_t(X_t|X_1), v_t^\theta(X_t)) \\ &= \mathbb{E}_{t, X_0, X_1 \sim \pi_{0,1}} D(\dot{\psi}_t(X_0|X_1), v_t^\theta(X_t)). \end{aligned} \tag{33}$$

**Remark B.9** *Unlike (25), the above training loss is feasible because $\psi_t(\cdot|x_1)$ is constructed, and $X_t = \psi_t(X_0|X_1)$ is known for each t. Although $\pi_{0,1}$ is unknown, $\pi_{\cdot|1}$ is constructed in the setup. Therefore, we can apply the Monte Carlo approximation*

$$\pi_{\cdot|1} \cdot \hat{q}^B \approx \pi_{\cdot|1} \cdot q = \pi_{0,1},$$

*where $\hat{q}^B$ is an $n$-size i.i.d. empirical distribution sampled from q.*

Conditional on $X_t = x$, the quantity $\dot{\psi}_t(X_0|X_1)$ is still a random variable, since multiple pairs $(X_0, X_1) = (x_0, x_1)$ may satisfy $\psi_t(x_0|x_1) = x$. That is, we aim to use a deterministic mapping $u^\theta(x)$ to approximate this random variable. As discussed in the previous section, the population solution of (33) is given by

$$u_t^*(x) = \mathbb{E}[\dot{\psi}_t(X_0|X_1) \mid X_t = x]. \tag{34}$$

At the end of this section, we introduce some classical examples of this model:

**Example B.10 (Song & Ermon (2019))** *In this work, the authors set $\pi_{0,1}(x_0, x_1) = p_0(x_0)p_1(x_1)$ (independent coupling), and define the interpolation as*

$$x_t = \psi_t(x_0|x_1) := x_1 + \sigma_t x_0, \tag{35}$$

*where $\sigma_t \in [0, 1]$ is a strictly monotone decreasing function with $\sigma_1 \approx 0$. The interpolation constraint (39) is thus slightly relaxed.*

*In this setting, we have*

$$p_t(x_t|x_1) = \mathcal{N}(x_t|x_1, \sigma_t^2 I_d),$$

$$v_t(x_t|x_1) := \dot{\sigma}_t x_0 = \frac{\dot{\sigma}_t}{\sigma_t}(x_1 - x_t),$$

$$\nabla \ln p_t(x_t|x_1) = -\frac{1}{\sigma_t^2}(x_t - x_1).$$

*Accordingly, the training loss is formulated as*

$$l(\theta; \sigma) := \mathbb{E}_{X_0, X_1 \sim \pi_{0,1},\, t \sim U[0,1]}\left[\left\|s_\theta(x_t, \sigma_t) + \frac{\tilde{x}-x}{\sigma_t^2}\right\|\right],$$

*where $\pi_{0,1} := \mathcal{N}(0, I_d) \otimes p_{\text{data}}$.*

*It is worth noting that in Song & Ermon (2019), the authors primarily use the* score function *formalism, and do not explicitly define the velocity field or interpolation function. However, their method can be naturally described within the flow matching framework, as discussed in Tong et al. (2023a); Lipman et al. (2024).*

**Example B.11 (Denoising Diffusion Probabilistic Model (DDPM), Ho et al. (2020))** *In this work, the authors use the independent coupling $\pi_{0,1} := \mathcal{N}(0, I_d) \otimes p_{data}$ and define the interpolation*

$$x_t = \phi_t(x_0|x_1) := \sqrt{\bar{\alpha}_t}\, x_1 + \sqrt{1 - \bar{\alpha}_t}\, x_0, \tag{36}$$

*where $\alpha_0 = 0,\ \alpha_1 = 1,\ \alpha_t \in [0, 1]$ (e.g., $\alpha_t = \sin(\frac{\pi}{2}t)$). The condition (32) is satisfied. Under this construction we have*

$$p_t(x_t|x_1) = \mathcal{N}(x_t|\sqrt{\bar{\alpha}_t}\, x_1, 1 - \bar{\alpha}_t),$$

$$v_t(x_t|x_1) = \dot{\alpha}_t x_1 - \frac{\alpha_t \dot{\alpha}_t}{\sqrt{1 - \alpha_t^2}} x_0 = \dot{\alpha}_t x_1 - \frac{\alpha_t \dot{\alpha}_t}{1 - \alpha_t^2} x_t,$$

$$\nabla \ln p_t(x_t|x_1) = -\frac{1}{1 - \alpha_t^2}(x_t - \alpha_t x_1).$$

*In the original paper, the interpolation is described as a discrete-time stochastic process. The authors derive*

$$p_{t-1|t,0}(x_{t-1}|x_t, x_0) = \mathcal{N}(x_{t-1}; \tilde{\mu}_t(x_t, x_0), \frac{1 - \bar{\alpha}_{t-1}}{1 - \bar{\alpha}_t}(1 - \alpha_t)),$$

$$\tilde{\mu}(x_t, t) = \frac{\sqrt{\bar{\alpha}_{t-1}}(1 - \alpha_t)}{1 - \bar{\alpha}_t}x_1 + \frac{\sqrt{\alpha_t}(1 - \bar{\alpha}_{t-1})}{1 - \bar{\alpha}_t}x_t$$

$$= \frac{1}{\sqrt{\alpha_t}}\left(x_t - \frac{1 - \alpha_t}{\sqrt{1 - \bar{\alpha}_t}}x_0\right).$$

*where $\alpha_t \in [0, 1]$ satisfies $\bar{\alpha}_t = \prod_{s \in [0,t]} \alpha_s$ in the discrete sense.*

*By introducing a parameterized mean*

$$\mu_\theta(x_t, t) := \frac{1}{\sqrt{\alpha_t}}\left(x_t - \frac{1 - \alpha_t}{1 - \bar{\alpha}_t}x_0^\theta(x_t, t)\right), \tag{37}$$

*matching $\mu_\theta(\cdot, \cdot)$ with $\tilde{\mu}(\cdot, \cdot)$ yields the loss function*

$$\mathbb{E}_{(X_0, X_1) \sim \pi_{0,1}}\left[\|X_0 - \epsilon^\theta(X_t, t)\|\right]. \tag{38}$$

*Since this model explicitly estimates $x_0$ (the Gaussian noise), it is known as the denoising diffusion model.*

**Example B.12 (Classical Conditional Flow Matching Lipman et al. (2023))** *In this work, the authors consider the independent coupling $\pi_{0,1} := \mathcal{N}(0, I_d) \otimes p_{data}$, and define the interpolation function as*

$$x_t = \phi_t(x_0|x_1) := tx_1 + (t\sigma_{\min} - t + 1)x_0,$$

*where $\sigma_{\min} \geq 0$ is a small constant. When $\sigma_{\min} = 0$, the constraint (32) is exactly satisfied. For $\sigma_{\min} > 0$, the final distribution $p_1$ becomes a Gaussian-perturbed version of $p_{data}$:*

$$p_1(x) = \int \mathcal{N}(x, \sigma_{\min}^2 I_d)\, dp_{data}(x_1) \approx p_{data}(x).$$

*The conditional distribution and velocity field are*

$$p_t(x_t|x_1) = \mathcal{N}(x_t|tx_1, (t\sigma_{\min} - t + 1)^2),$$

$$v_t(x_t|x_1) = x_1 + (\sigma_{\min} - 1)x_0 = x_1 + \frac{\sigma_{\min} - 1}{t\sigma_{\min} - t + 1}(x_t - tx_1).$$

*The training objective is then defined as*

$$\mathbb{E}_{X_0, X_1 \sim \pi_{0,1}}\left[\|X_1 - (1 - \sigma_{\min})X_0 - v^\theta(x_t, t)\|^2\right].$$

In this subsection, we consider the case where the conditioning variable is $Z = (X_0, X_1) = (x_0, x_1)$.

Similar to the previous section, the goal is to build a conditional probability path $p_{t|0,1}(\cdot|x_0, x_1)$ that satisfies the boundary conditions

$$p_{i|0,1}(x|x_0, x_1) = \delta_{x_i}(x), \quad \forall i \in \{0, 1\}. \tag{39}$$

We define a mapping $\psi : [0, 1] \times \mathbb{R}^d \times \mathbb{R}^d \to \mathbb{R}^d$ such that

$$\psi_t(x_0, x_1) = x_i, \quad \text{if } t = i, \; \forall i \in \{0, 1\}. \tag{40}$$

In Liu et al. (2022), $\psi_t$ is referred to as the **interpolation mapping**.

Let

$$p_{t|0,1}(\cdot|x_0, x_1) := \psi_t(\cdot, x_1)_\# \delta_{x_0}(\cdot) = \delta_{\psi_t(x_0, x_1)}(\cdot), \tag{41}$$

which by construction satisfies (39).

Define the random variable $X_t := \psi_t(X_0, X_1)$, whose marginal distribution is

$$p_t(\cdot) := \text{Law}(X_t) = \int p_{t|0,1}(\cdot|x_0, x_1) \, d\pi_{0,1}(x_0, x_1).$$

From Theorems B.1 and B.3, it follows that

$$v_t(x|x_0, x_1) := \dot{\psi}_t(x_0, x_1)$$

is the unique conditional velocity field that generates the conditional probability path $(p_{t|0,1}(\cdot|x_0, x_1))_{t \in [0,1]}$.

Thus, the conditional flow matching loss (29) reduces to

$$\mathcal{L}_{CFM}(\theta) := \mathbb{E}_{t,(X_0,X_1)\sim\pi_{0,1}, X_t \sim p_{\cdot|0,1}(\cdot|X_0,X_1)} \big[ D(v_t(X_t|X_0, X_1), v_t^\theta(X_t)) \big]$$

$$= \mathbb{E}_{t,(X_0,X_1)\sim\pi_{0,1}} \big[ D(\dot{\psi}_t(X_0, X_1), v_t^\theta(X_t)) \big]. \tag{42}$$

**Remark B.13** *Ignoring the difference in boundary conditions between $\psi_t(x_0|x_1)$ and $\psi_t(x_0, x_1)$, the training objectives (33) and (42) are essentially equivalent.*

**Example B.14 (Rectified Flow, Liu et al. (2022))** *The authors consider the independent coupling $\pi_{0,1}$ and define the interpolation*

$$x_t = \phi_t(x_0, x_1) = \alpha_t x_1 + \beta_t x_0,$$

*where $\alpha_0 = \beta_1 = 0$ and $\alpha_1 = \beta_0 = 1$, ensuring (40) is satisfied. The corresponding velocity field is*

$$v_t(x_t|x_0, x_1) = \dot{\alpha}_t x_1 + \dot{\beta}_t x_0.$$

*In the default choice $\alpha_t = t, \beta_t = 1 - t$, this simplifies to*

$$v_t(x_t|x_0, x_1) = x_1 - x_0,$$

*and the training loss becomes*

$$\mathbb{E}_{(X_0,X_1)\sim\pi_{0,1}} \big[ \|v_t^\theta(X_t|X_0, X_1) - (X_1 - X_0)\|^2 \big],$$

*a widely used formulation due to its simplicity and effectiveness.*

**Example B.15 (Stochastic Interpolation, Albergo et al. (2023))** *Here, randomness is introduced into the interpolation function. The stochastic interpolant is*

$$x_t = \phi_t(x_0, x_1, \xi) = (1 - t)x_0 + tx_1 + \sqrt{2t(1 - t)}\,\xi, \qquad t \in [0, 1],$$

*where $X_0 \sim \mathbf{p}$, $X_1 \sim \mathbf{q}$, and $\xi \sim \mathcal{N}(0, I_d)$ are independent.*

*Differentiating yields the velocity field*

$$v_t(x_t|x_0, x_1, \xi) = x_1 - x_0 + \frac{1 - 2t}{\sqrt{2t(1 - t)}}\,\xi.$$

*The training loss is*

$$\mathcal{L}_{\text{SI}}(\theta) = \mathbb{E}_{(X_0,X_1)\sim\boldsymbol{\pi}_{0,1}, \xi\sim\mathcal{N}(0,I_d)} \big[ \|v_t^\theta(X_t|X_0, X_1, \xi) - v_t(X_t|X_0, X_1, \xi)\|^2 \big],$$

*where $X_t = \phi_t(X_0, X_1, \xi)$.*

*This reduces to rectified flow when the noise vanishes ($\xi = 0$). For intermediate $t$, the stochastic term encourages the model to learn a velocity field that balances interpolation with diffusion-like dynamics, effectively bridging flow matching and score-based diffusion models.*

**Example B.16 (Independent Conditional Flow Matching, Lipman et al. (2023))** *The   method discussed in Example B.12 can also be described in the setting $Z = (X_0, X_1)$. In this case, the interpolation function is*

$$I_t(x_0, x_1, \xi) = (1 - t)x_0 + tx_1 + \sigma\xi, \qquad t \in [0, 1],$$

with independent coupling $\boldsymbol{\pi}_{0,1} = \mathbf{p} \otimes \mathbf{q}$. Note that under this formulation, the source distribution becomes $\mathbf{p}_0 = \mathbf{p} * \mathcal{N}(0, I_d)$ (where $*$ denotes convolution), and the target distribution becomes $\mathbf{p}_1 = \mathbf{q} * \mathcal{N}(0, I_d)$.

The corresponding conditional velocity field is

$$v_t(x_t \mid x_0, x_1) = \frac{d}{dt} \mathbb{E}_\xi [I_t(x_0, x_1, \xi)] = x_1 - x_0.$$

Thus, the training loss is

$$\mathcal{L}_{\mathrm{CFM}}(\theta) := \mathbb{E}_{(X_0, X_1) \sim \mathbf{p} \otimes \mathbf{q}} \, \mathbb{E}_{t \sim U[0,1], \, \xi \sim \mathcal{N}(0, I_d)} \big[ \| v_t^\theta(X_t) - (X_1 - X_0) \|^2 \big], \tag{43}$$

where $X_t = (1 - t)X_0 + tX_1 + \sigma\xi$.

Because of its simplicity and effectiveness, Independent CFM has become one of the most widely used training objectives for flow-based generative models.

## B.3 OT-BASED FLOW MATCHING MODELS

When we consider $Z = (X_0, X_1)$, a natural extension of the above flow matching models is to utilize optimal transport to define $\pi_{0,1}$.

**Example B.17 (Mini-Batch OT Flow, Pooladian et al. (2023))** *A classical approach is Mini-Batch Optimal Transport. Here, we sample i.i.d. empirical distributions $\mathbf{p}_0^B, \mathbf{p}_1^B$ from $\mathbf{p}_0 = \mathbf{p}$ and $\mathbf{p}_1 = \mathbf{q}$, respectively, with batch size $B \in \mathbb{N}$. Let $\boldsymbol{\pi}^*(\mathbf{p}_0^B, \mathbf{p}_1^B)$ denote the optimal transportation plan between $\mathbf{p}_0^B$ and $\mathbf{p}_1^B$. This empirical coupling is then used during training as a proxy for the true coupling between $\mathbf{p}$ and $\mathbf{q}$. Formally, the training objective is*

$$\mathcal{L}_{\mathrm{OT\text{-}CFM}}(\theta) := \mathbb{E}_{\substack{X_0^B \sim i.i.d. \; \mathbf{p}, \\ X_1^B \sim i.i.d. \; \mathbf{q}}} \mathbb{E}_{(X_0, X_1) \sim \pi_{0,1}} \big[ \| v_t^\theta(X_t) - (X_1 - X_0) \|^2 \big], \tag{44}$$

*where $\boldsymbol{\pi}_{0,1}$ is the optimal coupling in $OT(\mathbf{p}^B, \mathbf{q}^B)$ with empirical laws*

$$\mathbf{p}^B = \mathrm{Law}(X_0^B), \qquad \mathbf{q}^B = \mathrm{Law}(X_1^B). \tag{45}$$

*Pooladian et al. (2023) show that the transportation cost (trajectory length) induced by the mini-batch OT plan is strictly smaller than that of the independent coupling. This provides a theoretical justification for why OT-based conditional flow matching yields more cost-efficient and geometrically faithful interpolations.*

**Example B.18 (Mini-Batch OT and Sinkhorn OT Stochastic Flow)** *In Tong et al. (2023b), the authors combine the OT-CFM model (44) with the stochastic conditional flow matching model (43). The training loss is*

$$\mathcal{L}_{\mathrm{OT\text{-}CFM}}(\theta) := \mathbb{E}_{\substack{X_0^B \sim \mathbf{p} \\ X_1^B \sim \mathbf{q}}} \mathbb{E}_{\substack{(X_0, X_1) \sim \pi_{0,1} \\ t, \xi \sim \mathcal{N}(0, I_d)}} \big[ \| v_t^\theta(X_t) - (X_1 - X_0) \|^2 \big],$$

*with interpolation*

$$X_t = I_t(X_0, X_1, \xi) = (1 - t)X_0 + tX_1 + \sigma\xi, \tag{46}$$

*where $\boldsymbol{\pi}_{0,1}$ is the optimal solution of the mini-batch OT problem.*

*Compared to independent coupling, the OT-induced coupling aligns the source and target samples in a globally optimal way, producing straighter transport trajectories and reducing unnecessary curvature in the learned flows. This leads to more stable training and improved sample efficiency.*

*The authors further consider the entropic OT solution for $\boldsymbol{\pi}_{0,1}$, leading to the* Schrödinger Bridge CFM *model:*

$$\mathcal{L}_{\mathrm{SB\text{-}CFM}}(\theta) := \mathbb{E}_{\substack{X_0^B \sim \mathbf{p} \\ X_1^B \sim \mathbf{q}}} \mathbb{E}_{\substack{(X_0, X_1) \sim \pi_{0,1} \\ t, \xi \sim \mathcal{N}(0, I_d)}} \big[ \| v_t^\theta(X_t) - v_t(X_t | X_0, X_1) \|^2 \big],$$

*with interpolation and velocity field*

$$X_t = (1-t)X_0 + tX_1 + \sqrt{t(1-t)}\sigma\xi, \tag{47}$$

$$v_t(x|x_0, x_1) = \frac{(1-2t)}{2t(1-t)}(x - \bar{x}_t) + (x_1 - x_0), \qquad \bar{x}_t = (1-t)x_0 + tx_1, \tag{48}$$

$$\boldsymbol{\pi}_{0,1} \text{ is optimal for } OT_{2\sigma^2}(\mathrm{Law}(X_0^B), \mathrm{Law}(X_1^B)).$$

*Here, entropic OT regularization further smooths the coupling, interpolating between deterministic OT alignments and independent couplings, thereby improving robustness.*

**Example B.19 (Optimal Flow Matching (OFM) Kornilov et al. (2024))** *This method modifies the flow matching framework by restricting the velocity fields to gradients of convex potentials. Concretely, the authors parameterize $\psi$ with an Input Convex Neural Network (ICNN) and define $v(x) = \nabla\psi(x)$.*

*We first recall the Kantorovich dual formulation of quadratic optimal transport. For two probability measures $\mathbf{p}$ and $\mathbf{q}$ on $\mathbb{R}^d$, the squared 2-Wasserstein distance admits the following dual form:*

$$OT(\mathbf{p}, \mathbf{q}) = \mathbb{E}_{X_0 \sim \mathbf{p}}\|X_0\|^2 + \mathbb{E}_{X_1 \sim \mathbf{q}}\|X_1\|^2 - 2\sup_{\psi \text{ convex}}\left\{\mathbb{E}_{X_0 \sim \mathbf{p}}\psi(X_0) + \mathbb{E}_{X_1 \sim \mathbf{q}}\psi^*(X_1)\right\}, \tag{49}$$

*where $\psi$ is any convex function and $\psi^*$ is its convex conjugate (Villani, 2003; Benamou & Brenier, 2000). Brenier's theorem ensures that the Monge optimal map under quadratic cost is of the form $T^\star = \nabla\psi^\star$, and the optimal velocity field in (Benamou–Brenier) is $\nabla\psi^\star(x) - x$, where $\psi^\star$ is the maximizer in (49).*

**OFM model.** *Given a coupling $\boldsymbol{\pi}$ between $\mathbf{p}$ and $\mathbf{q}$, samples $(x_0, x_1) \sim \boldsymbol{\pi}_{0,1}$, and interpolation $x_t = (1-t)x_0 + tx_1$, the OFM objective is*

$$\mathcal{L}_{\mathrm{OFM}}(\psi) = \mathbb{E}\Big[\|u^\psi(x_t) - (x_1 - x_0)\|^2\Big],$$

$$u^\psi(x_t) = \nabla\psi(x_t) - x_0, \quad \psi \text{ convex}.$$

*At the population optimum, minimizing this objective recovers the Brenier map $\nabla\psi^\star$; equivalently, $\psi^\star$ solves the dual Kantorovich problem. This aligns flow matching with the dual OT formulation and guarantees straight displacement interpolations.*

*Intuitively, unlike standard FM/CFM models, the mapping $x \mapsto \psi(x)$ (or $x \mapsto \nabla\psi(x)$) does not take time $t$ as input. This is because the optimal velocity field in the OT problem has constant speed. OFM exploits this property to simplify the model while preserving optimality.*

## C EXPERIMENT SETTING DETAILS IN IMAGE GENERATION.

We study one-step MeanFlow generation on the MNIST dataset, operating entirely in the latent space of a pretrained VAE tokenizer from Rombach et al. (2022). Each $28\times28$ grayscale digit is padded to $32\times32$, normalized to [-1,1], and replicated across three channels before being encoded once by the frozen VAE. The resulting $4\times4\times4$ latents are cached and reused throughout training. Our generator adopts a ConvNeXt-style U-Net backbone, following the implementation from Geng et al. (2025), but adapted to the low-resolution latent tensor ($\approx 59M$ parameters). We retain dual sinusoidal embeddings for the flow time $t$ and the solver step size $h$, such that the network is explicitly conditioned on both temporal signals, consistent with the original design.

Training hyperparameters largely mirror the baseline from Geng et al. (2025) with minor modifications to improve latent-space stability. We use Adam with a learning rate of $1 \times 10^{-3}$, $(\beta_1, \beta_2) = (0.9, 0.99)$, batch size 256, no weight decay, and 30k iterations with a 10% linear warmup followed by a constant schedule. Exponential moving averages are maintained with decay 0.99 and an update period of 16 steps. Timesteps are sampled from a logit-normal distribution with $(P_{\mathrm{mean}}t = -0.6, P_{\mathrm{std}}t = 1.6, P_{\mathrm{mean}}r = -4.0)$ and a mismatch ratio of 0.75. The training objective follows the JVP-based loss with adaptive reweighting as introduced in Geng et al. (2025). Beyond the default Gaussian pairing, we also evaluate transport-based pairings, including Optimal Transport (OT), Sinkhorn OT, Low-Rank Linear OT, hi er OT, and Sliced OT.

We evaluate several typical OT solvers for Mean Flows by varying the Number of Function Evaluations (NFE) under the Euler integrator. By default we study 1–NFE generation and report results up to 10–NFE. Evaluation metrics use Fréchet Inception Distance (FID) Heusel et al. (2017) and the 2–Wasserstein distance ($W_2$) Villani et al. (2008), computed between generated images and reconstructed images from VAE. In pixel space, we use the Inception network from `TorchMetrics` to compute FID and $W_2$. In latent space, we re-encode generated and reconstructed images with the frozen VAE to obtain $4 \times 4 \times 4$ latent vectors (rescaled by $0.18215$) and report $\text{FID}^{\text{ae}}$ and $W_2^{\text{ae}}$. While FID captures perceptual quality and diversity in Inception feature space, the $W_2$ provides a more direct measure of distribution alignment between generated and VAE-reconstructed samples, both in pixel space and autoencoder latent manifold.

**One-step generation results.** Table 4 reports one-step (1–NFE) MNIST generation for baseline and six transport-based pairings sampler. The left sub-table (a) uses EMA parameters and the right panel (b) uses original weights. We find the exact OT pairing has the lowest scores on all four metrics, while LOT-HR and LOT-LR are competitive under EMA. The trends of $W_2$, $\text{FID}^{\text{ae}}$, and $W_2^{\text{ae}}$ mirror those of FID, indicating reduced divergence between generated and reference distributions in both pixel and latent spaces when transport-based pairings are applied.

Table 4: One-step generation performance on MNIST. FID, $W_2$ and $\text{FID}^{\text{ae}}$, $W_2^{\text{ae}}$ are computed between generated images and reconstructed images from VAR. Best values are bold with gray background.

| (a) EMA=True | | | | | (b) EMA=False | | | | |
|---|---|---|---|---|---|---|---|---|---|
| Method | FID ↓ | $W_2$ ↓ | $\text{FID}^{\text{ae}}$ ↓ | $W_2^{\text{ae}}$ ↓ | Method | FID ↓ | $W_2$ ↓ | $\text{FID}^{\text{ae}}$ ↓ | $W_2^{\text{ae}}$ ↓ |
| w/o OT (Gaussian) | 3.6709 | 8.7449 | 0.2296 | 2.5706 | w/o OT (Gaussian) | 8.0620 | 9.2343 | 0.3792 | 2.6661 |
| OT | **1.9179** | **8.2102** | **0.0304** | **2.3527** | OT | **3.2484** | **8.6393** | **0.0993** | **2.4512** |
| LOT-LR | 2.2258 | 8.4315 | 0.0405 | 2.3751 | LOT-LR | 6.2175 | 8.9047 | 0.1397 | 2.4792 |
| LOT-HR | 1.9754 | 8.3815 | 0.0401 | 2.3557 | LOT-HR | 5.5294 | 8.8883 | 0.1077 | 2.4966 |
| Sinkhorn | 3.6944 | 8.6554 | 0.1672 | 2.5144 | Sinkhorn | 9.9643 | 9.6867 | 0.3419 | 2.7312 |
| Sliced-OT | 7.2018 | 9.1260 | 0.9254 | 2.8637 | Sliced-OT | 11.2034 | 9.7436 | 0.8810 | 2.9050 |
| OT-Partial | 4.1926 | 8.9223 | 0.2917 | 2.6078 | OT-Partial | 8.3549 | 9.4912 | 0.3822 | 2.6973 |

**Multi-step generation results.** From Table 5 we find in pixel space, the exact OT solver attains the lowest FID and $W_2$ at all 2/5/10 NFEs. Within the autoencoder manifold, $\text{FID}^{\text{ae}}$ is lowest for LOT-LR across steps, while $W_2^{\text{ae}}$ alternates between OT (NFE=2,10) and LOT-LR (NFE=5). Improvements from NFE=2 to 5 are significant, whereas gains from 5 to 10 are small, suggesting diminishing returns beyond 5 steps.

**Multi-step generation trends across NFEs.** Figure 6 plots FID, $W_2$, $\text{FID}^{\text{ae}}$, and $W_2^{\text{ae}}$ versus NFE (1–10) with EMA. The curves validate Table 5 that OT dominates in pixel-space metrics across steps, LOT-LR leads on $\text{FID}^{\text{ae}}$, and $W_2^{\text{ae}}$ is shared between OT and LOT-LR. Most OT Solvers improve rapidly up to $\sim$5 NFE, after which the curves flatten and the solver rankings remain stable.

# D EXPERIMENT SETUP DETAILS IN CONDITIONAL SHAPE GENERATION

## D.1 MEAN FLOW MATCHING UNDER GUIDANCE

We first recap the mean flow matching model with guidance.

Following the convention in the mean flow formulation Geng et al. (2025), we set the condition $Z = (X_0, X_1)$. Guidance is represented by a random variable $\mathbf{c}$ such that $(x_{\text{data}}, \mathbf{c})$ follows a joint distribution. For example, $\mathbf{c}$ may correspond to the class label or extracted features of $x_{\text{data}}$.

In the classical flow matching setting (see, e.g., Lipman et al. (2024)), the guided ground-truth velocity field is defined as

$$v_t^{\text{cfg}}(x_t \mid \mathbf{c}) = \omega v_t(x_t \mid \mathbf{c}) + (1 - \omega) v_t(x_t), \tag{50}$$

Table 5: Multi-step generation performance on MNIST (EMA=True) at 2/5/10–NFEs. metrics are computed between generated images and reconstructed images from VAE, best values per column are bold with gray background.

| Method | FID ($\downarrow$) | | | $W_2$ ($\downarrow$) | | | FID$^{\text{ae}}$ ($\downarrow$) | | | $W_2^{\text{ae}}$ ($\downarrow$) | | |
|---|---|---|---|---|---|---|---|---|---|---|---|---|
| | 2 | 5 | 10 | 2 | 5 | 10 | 2 | 5 | 10 | 2 | 5 | 10 |
| w/o OT (Gaussian) | 1.0880 | 0.6318 | 0.7267 | 8.2560 | 8.0634 | 8.0602 | 0.2878 | 0.1860 | 0.1811 | 2.4788 | 2.3719 | 2.3639 |
| OT | **0.6123** | **0.4689** | **0.4935** | **8.0383** | **8.0029** | **7.9546** | 0.0484 | 0.0621 | 0.0644 | **2.3273** | 2.3089 | **2.2885** |
| LOT-LR | 0.7449 | 0.5371 | 0.5531 | 8.1481 | 8.0312 | 7.9941 | **0.0439** | **0.0536** | **0.0615** | 2.3331 | **2.2975** | 2.2921 |
| LOT-HR | 0.8357 | 0.6053 | 0.5759 | 8.1922 | 8.0683 | 8.0138 | 0.0496 | 0.0577 | 0.0631 | 2.3457 | 2.3073 | 2.2990 |
| Sinkhorn | 1.0782 | 0.6362 | 0.7135 | 8.2903 | 8.1000 | 8.1040 | 0.2644 | 0.1598 | 0.1641 | 2.4663 | 2.3700 | 2.3629 |
| Sliced-OT | 2.9616 | 1.2293 | 0.9985 | 8.4144 | 8.1957 | 8.1968 | 0.3852 | 0.2581 | 0.2806 | 2.5695 | 2.4584 | 2.4733 |
| OT-Partial | 2.9925 | 2.9214 | 2.8793 | 8.6410 | 8.5239 | 8.5422 | 0.3711 | 0.4142 | 0.4340 | 2.6095 | 2.5879 | 2.6077 |

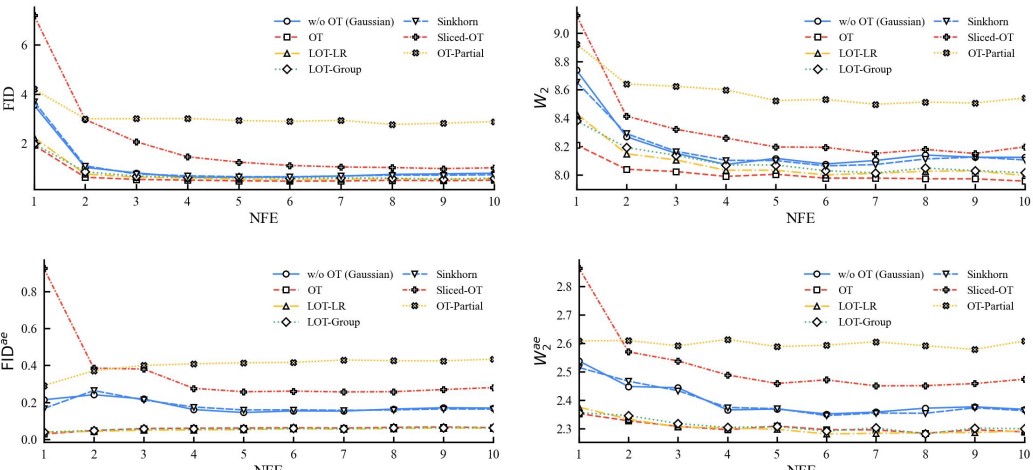

Figure 6: Multi-step generation performance results on MNIST (EMA=True). panels (a–d) plot FID, $W_2$, FID$^{\text{ae}}$, and $W_2^{\text{ae}}$ versus NFE (1–10).

where $\omega \geq 1$ is the **guidance scale**. Here $v_t(x_t)$ and $v_t(x_t \mid \mathbf{c})$ denote the marginal velocity fields based on $p_t$ and $p_{t|\mathbf{c}}$, respectively:

$$v_t(x) = \mathbb{E}_{(X_0,X_1)\sim\boldsymbol{\pi}_{0,1},\, X_t\sim\mathbf{p}_{t|X_0,X_1}} \left[ \tfrac{d}{dt} I_t(X_0,X_1) \right] = \mathbb{E}_{(X_0,X_1)\sim\boldsymbol{\pi}_{0,1}}[X_1 - X_0],$$

$$v_t(x \mid \mathbf{c}) = \mathbb{E}_{(X_0,X_1)\sim\boldsymbol{\pi}_{0,1}|\mathbf{c},\, X_t\sim\mathbf{p}_{t|X_0,X_1,\mathbf{c}}} \left[ \tfrac{d}{dt} I_t(X_0,X_1) \right] = \mathbb{E}_{(X_0,X_1)\sim\boldsymbol{\pi}_{0,1}|\mathbf{c}}[X_1 - X_0].$$

In both cases, the second equality holds under the deterministic interpolation $I_t(x_0,x_1) = (1 - t)x_0 + tx_1$. Indeed, in this setting $p_{t|x_0,x_1} = p_{t|x_0,x_1,\mathbf{c}} = \delta_{(1-t)x_0+tx_1}$, and $\frac{d}{dt} I_t(x_0,x_1) = x_1 - x_0$.

Based on Geng et al. (2025), the guided mean velocity is defined as

$$u^{\text{cfg}}(x_t, r, t \mid \mathbf{c}) = \frac{1}{t - r} \int_r^t v^{\text{cfg}}(\tau, x_\tau \mid \mathbf{c}) \, d\tau.$$

Multiplying both sides by $(t - r)$ and differentiating with respect to $t$, we obtain

$$u^{\text{cfg}}(x_t, t, r \mid \mathbf{c}) = v^{\text{cfg}}(\tau, x_\tau \mid \mathbf{c}) - (t - r)\frac{d}{dt} u^{\text{cfg}}(t, r, x_t \mid \mathbf{c})$$

$$= v^{\text{cfg}}(\tau, x_\tau \mid \mathbf{c}) - (t - r)\big(v_t^{\text{cfg}}(x_t \mid \mathbf{c})\, \partial_{x_t} u^{\text{cfg}} + \partial_t u^{\text{cfg}}\big).$$

Moreover, we have the identity

$$v^{\text{cfg}}(t, x_t \mid \mathbf{c}) = \omega v(t, x_t \mid \mathbf{c}) + (1 - \omega)v(t, x_t)$$

$$= \omega v(t, x_t \mid \mathbf{c}) + (1 - \omega)v^{\text{cfg}}(t, x_t)$$

$$= \omega v(t, x_t \mid \mathbf{c}) + (1 - \omega)u^{\text{cfg}}(t, t, x_t),$$

where $v^{\mathrm{cfg}}(t, x_t) := \mathbb{E}_{\mathbf{c}}[v^{\mathrm{cfg}}(t, x_t \mid \mathbf{c})]$, and

$$u^{\mathrm{cfg}}(t, r, x_t) := \mathbb{E}_{\mathbf{c}}[u^{\mathrm{cfg}}(t, r, x_t \mid \mathbf{c})] = u^{\mathrm{cfg}}(t, r, x_t \mid \emptyset),$$

with $\emptyset$ denoting the unconditional case.

Combining the above identities, Geng et al. (2025) introduces the training loss for mean flow with guidance:

$$\mathcal{L}(\theta) = \mathbb{E}\big[\|u_\theta^{\mathrm{cfg}}(t, r, x_t \mid \mathbf{c}) - \mathrm{sg}(u_{\mathrm{tgt}})\|^2\big],$$

$$u_{\mathrm{tgt}} := \tilde{v}_t - (t - r)\big(\tilde{v}_t \, \partial_z u_\theta^{\mathrm{cfg}} + \partial_t u_\theta^{\mathrm{cfg}}\big),$$

$$\tilde{v}_t := \omega v_t + (1 - \omega)u_\theta^{\mathrm{cfg}}(t, t, x_t).$$

Based on this process, we can derive the training loss of OT-MeanFlow under guidance as

$$\mathbb{E}_{\mathbf{c} \sim (1-\eta)\mathbf{p_c} + \eta\delta_\emptyset} \; \mathbb{E}_{\substack{X_0^B \sim \mathbf{p} \\ X_1^B \sim \mathbf{q} \mid \mathbf{c}}} \; \mathbb{E}_{(X_0, X_1) \sim \pi_{0,1}^{\mathbf{c}}} \left[ \|u_\theta^{\mathrm{cfg}}(t, r, x_t \mid \mathbf{c}) - u_{\mathrm{tgt}}\|^2 \right], \tag{51}$$

where $\pi_{0,1}^{\mathbf{c}}$ denotes the optimal coupling between $\mathrm{Law}(X_0^B)$ and $\mathrm{Law}(X_1^B)$. We use the superscript $\mathbf{c}$ to emphasize that $X_1^B$ is sampled from the conditional distribution $\mathbf{q} \mid \mathbf{c}$. During the experiment, we set $\eta = 0$.

## D.2 EXPERIMENTAL DETAILS ON CONDITIONAL SHAPE GENERATION

**Training and evaluation**   We pre-train a PointNet-based auto-encoder with two additional linear layers, followed by batch normalization and max pooling for the encoder. We minimize the Chamfer distance between the reconstructed shape, and the ground truth. The number of epochs is set to 1000. We train an auto-encoder on ShapeNet Chairs, and one on ModelNet10, and utilize these pretrained auto-encoders to extract context features as condition vectors to our generation model.

For training the MeanFlow model, the context vector extracted from the pre-trained auto-encoder is first then projected through a two layer MLP with an output size of 256. This is then concatenated alongside the flow model input and a Residual MLP network is used for flow prediction. This model has 12 layers and hidden dimension set to 2048. For ShapeNet Chairs, use the train-validation split from class "Chairs" to train the model, and report the evaluation metrics and plots on the test set. Similarly, for ModelNet10, we train on the training split and report evaluation metrics on the test split.

All experiments are trained for 1000 epochs. We train across 4 NVIDIA A6000 GPUs with a batch size of 32 graphs. For each graph, we then randomly sample 256 points as target samples. We use the Adam optimizer with $lr = 2e - 5$. The source distribution is a randomly generated gaussian with the same dimensionality as the target data.

**Interpolation**   For the interpolation plots, we condition the model on a convex combination of context features for two random shapes. Assume $C_1$ and $C_2$ are context vectors for shape 1 and 2 respectively. The combined context vector is formulated as $(1 - \alpha)C_1 + \alpha C_2$. Ideally, the output conditioned on this context vector should display an interpolated version of the two shapes. Additional interpolation results for LOT-ind and LOT-group are provided in Figure 7. It can be observed that other OT variants also preserve a good performance, capturing finer details compared to MF.

## D.3 EXPERIMENTAL DETAILS ON UNPAIRED IMAGE-TO-IMAGE TRANSLATION

We use a 4-layer MLP with hidden dimension of 1024. The time inputs $t$ and $h$ are concatenated and projected to a 32-dimensional vector through an MLP layer. We use the Adam optimizer with a learning rate $1e - 3$, and train all methods for 5000 epochs with a batch size of 2048.

## E   FUTURE DIRECTION.

One of our future directions is applying generalized sliced OT into the old (flow matching) and new (mean flow matching) methods.

Figure 7: Single step (NEF=1) shape interpolation on two samples from *ShapeNet* chairs using LOT-LR, and LOT-HR.

### E.1 BACKGROUND: SLICED OPTIMAL TRANSPORT (SOT).

Sliced OT (Rabin et al., 2011; Bonneel et al., 2015) reduces high-dimensional OT to a collection of one-dimensional OT problems, which admit closed-form solutions. For a probability measure $\mathbf{p}$ on $\mathbb{R}^d$ and a projection direction $\theta \in \mathbb{S}^{d-1}$, let $\mathcal{R}_\theta$ denote the Radon transform (i.e., 1D projection). The sliced OT distance is defined as

$$\text{SOT}^2(\mathbf{p}, \mathbf{q}) = \int_{\mathbb{S}^{d-1}} OT(\mathcal{R}_\theta \mathbf{p}, \mathcal{R}_\theta \mathbf{q}) \, d\theta, \tag{52}$$

where the 1D Wasserstein distances can be computed in $\mathcal{O}(n \log n)$ via sorting. In practice, the integral is approximated by Monte Carlo sampling over random directions.

**Generalized Radon Transform.** In the simplest setting, the Radon transform uses the inner product as the 1D projection:

$$\mathcal{R}_\theta \mathbf{p} := \langle \theta, \cdot \rangle_{\#} \mathbf{p}.$$

Later, this transform was generalized to nonlinear mappings:

$$G\mathcal{R}_\theta \mathbf{p} := \langle \theta, h(\cdot) \rangle_{\#} \mathbf{p},$$

where $h : \mathbb{R}^d \to \mathbb{R}^{d'}$ satisfies certain regularity conditions and can be modeled as a learnable neural network. Intuitively, $h$ serves as a feature mapping into a Reproducing Kernel Hilbert Space (RKHS), and the inner-product projection is then computed in the transformed space.

**Sliced OT Plan.** Let $\gamma^\theta$ denote the optimal plan for the 1D OT problem. One can lift $\gamma^\theta$ back into $\mathbb{R}^d$ (see, e.g., Mahey et al. (2023); Liu et al. (2024)), denoted as $\mathcal{L}(\gamma^\theta)$. In the discrete case, $\gamma^\theta$ is represented as an $n \times m$ transport matrix; with probability 1, $\gamma^\theta$ and $\mathcal{L}(\gamma^\theta)$ coincide. Therefore, for convenience, we do not distinguish between them in this article.

There are several ways to define a transportation plan between $\mathbf{p}$ and $\mathbf{q}$ in the sliced OT setting, for example:

$$\begin{cases} \gamma_{\text{SOT-min}} := \arg\min_{\gamma^\theta} \langle C, \gamma^\theta \rangle, \\ \gamma_{\text{SOT-expect}} := \mathbb{E}_{\theta \sim \text{Unif}(\mathbb{S}^{d-1})}[\gamma^\theta], \\ \gamma_{\text{SOT-temp}} := \mathbb{E}_{\theta \sim \text{Unif}(\mathbb{S}^{d-1})}\left[\gamma^\theta \frac{\exp(-\lambda\langle C, \gamma^\theta \rangle)}{\int_{\mathbb{S}^{d-1}} \exp(-\lambda\langle C, \gamma^{\theta'} \rangle) \, d\theta'}\right], \end{cases} \tag{53}$$

where in the third case, $\lambda \geq 0$ controls the temperature.

In practice, these expectations are approximated via Monte Carlo sampling over projection directions $\theta$.

**Differentiable transportation plan** One challenge of the above formulations is the minimization over $\theta$. Classical gradient descent does not work since $\gamma^\theta$ is not differentiable with respect to $\theta$. To address this issue, several techniques have been proposed.

- The simplest method is **Soft-sorting** (Prillo & Eisenschlos, 2020). When $\mathcal{G}\mathcal{R}\mathbf{p} = \frac{1}{B}\sum_{i=1}^{B} \delta_{x_i}$ and $\mathcal{G}\mathcal{R}\mathbf{q} = \frac{1}{B}\sum_{j=1}^{B} \delta_{y_j}$ are empirical distributions on $\mathbb{R}$ with equal weights,

the optimal coupling for any convex cost $c(x, y) = |x - y|^p$ $(p \geq 1)$ matches points in *sorted order*. Let $S_x, S_y \in \{0, 1\}^{B \times B}$ be permutation matrices such that $S_x \mathbf{x} = \mathbf{x}^\uparrow$ and $S_y \mathbf{y} = \mathbf{y}^\uparrow$, where $\mathbf{x} = [x_1, \ldots, x_B]^\top$, $\mathbf{y} = [y_1, \ldots, y_B]^\top$, and $^\uparrow$ denotes nondecreasing sort. Then the optimal plan is

$$\gamma^* = \frac{1}{B} S^\top(\mathbf{x}) S(\mathbf{y}).$$

Intuitively, it denotes the matching

$$x_i^\uparrow \mapsto y_i^\uparrow, \forall i \in [1 : B].$$

Inspired by this formulation, we replace the hard sorting $S_x, S_y$ by the corresponded *soft permutation* matrices $S_\tau(\mathbf{x}), S_\tau(\mathbf{y}) \in \mathcal{B}$ (doubly-stochastic), yielding the relaxed plan

$$\gamma_\tau = \frac{1}{B} S_\tau(\mathbf{x})^\top S_\tau(\mathbf{y}),$$

$$S_{\tau, \mathbf{x}} = \text{softmax}(-d(\text{sort}(\mathbf{x})1^\top - \mathbf{x}^\top 1)/\tau)$$

which recovers the hard coupling as $\tau \to 0$. $S_\tau(\mathbf{x}), S_\tau(\mathbf{y})$ are differentiable when $\tau > 0$, thus we obtain a differentiable plan.

- The second method to obtain a differentiable plan is **Gaussian perturbation**. We define the smoothed objective

$$h_\varepsilon(\theta) = \mathbb{E}_{Z \sim \mathcal{N}(0, I)}[h(\theta + \varepsilon Z)],$$

$$\text{where } h(\theta) = OT(\mathcal{GR}_\theta(\mathbf{p}), \mathcal{GR}_\theta(\mathbf{q})).$$

By Stein's lemma, the gradient of the smoothed objective admits the unbiased form

$$\nabla_\theta h_\varepsilon(\theta) = \frac{1}{\varepsilon} \mathbb{E}_{Z \sim \mathcal{N}(0, I)}[h(\theta + \varepsilon Z) Z]. \tag{54}$$

In practice, we approximate the expectation using Monte Carlo with a control variate, leading to the empirical estimator

$$\widehat{\nabla_\theta h_\varepsilon}(\theta) = \frac{1}{\varepsilon N} \sum_{k=1}^{N} \left( h(\theta + \varepsilon z_k) - h(\theta) \right) z_k, \qquad z_k \sim \mathcal{N}(0, I). \tag{55}$$

This yields a differentiable surrogate for the originally non-smooth transport objective.

### E.2 FUTURE WORK: SLICED OT MEAN FLOW

One natural extension of the proposed OT-mean flow method is utilizing the sliced OT plan 53 to define $\pi_{0,1}$ in the mean flow (or original flow). The current challenges include the following:

- In a high-dimensional data generation experiment, it is important to define a suitable generalized Radon transform $\mathcal{GR}$ as we aim to capture the important features in the high-dimensional original space. How to efficiently train such a feature mapping $h(x)$ is still unclear.

- Due to the nature of the sliced OT problem, the mapping $\theta \mapsto OT(\mathcal{GR}_\theta(\mathbf{p}), \mathcal{GR}_\theta(\mathbf{q}))$ is not differentiable at finite points. It is still unclear if the gradient-descent based optimization method is suitable.

- The number of projections required to achieve an accurate sliced approximation scales with data complexity; balancing computational efficiency and approximation quality is still an open question.

- It remains unclear how to integrate sliced OT plans with stochastic mini-batch training while preserving stability and convergence guarantees in mean flow training.

Despite these challenges, combining sliced OT with mean flow has significant potential benefits. By working with one-dimensional projections, sliced OT can substantially reduce computational complexity compared to solving high-dimensional OT directly. Moreover, if an effective feature mapping $\mathcal{GR}$ can be learned, this framework could also adaptively emphasize task-relevant directions in the data, leading to improved sample quality and representation learning.

# F COMPUTATIONAL RESOURCE

The toy shape translation experiments were conducted on an AMD EPYC 7713 CPU.

The image generation experiments were conducted on a single NVIDIA A6000 GPU with 48 GB memory.

The unpaired image-to-image translation experiments were trained on a single NVIDIA A6000 GPU with 48 GB memory. The point cloud experiments were done using distributed training, parallelized over $4\times$ NVIDIA A6000 GPUs.

