# OpenReview forum: "TRANSPORT-BASED MEAN FLOWS FOR GENERATIVE MODELING"
_ICLR.cc/2026/Conference — ICLR 2026 Conference Desk Rejected Submission_

### Official Review · Reviewer_3Yge · 2025-10-15

**Soundness:** 3
**Presentation:** 3
**Contribution:** 2
**Rating:** 4
**Confidence:** 4

**Summary:**

The paper proposes to improve the training of Mean Flow models by incorporating optimal transport. Specifically, minibatch OT methods (i.e. computing OT on the empirical distribution represented by the minibatch) define a coupling on the elements of a batch that is used to train the models. Intuitively, this results in straighter trajectories that should be simpler to learn with Mean Flows. Experiments on toy data, MNIST, shapes, and paired image-to-image translation are presented.

**Strengths:**

- Clear presentation of mathematics and motivation
- Good overview of relation to previous work
- Proposes a natural - yet novel - idea.
- A diverse set of experiments is performed and some improvements are shown

**Weaknesses:**

- While a natural idea, the method has limited novelty.
- The experiments are limited: MNIST is still a toy data and it is only performed on a latent space of size 4x4x4. All improvements are paid for by significant longer training times per epoch. Therefore, it is unclear how much the use of the OT coupling is really useful here.

**Questions:**

Typos/comments:
- Abstract, first word: Flow-matching -> Flow matching
- Page 3: “and introduces stochasticity into the model”. It does not introduce further stochasticity and I would remove that comment.

Missing references:
- Introduction: Mean Flow is equivalent to Flow Map Matching [1] and they should be cited together

[1] https://arxiv.org/abs/2406.07507

---

> ### Author Response · Authors · 2025-12-03
> **Response to Reviewer 3Yge**
>
> We sincerely thank the reviewer for their encouraging comments on the clarity of the presentation, the motivation, and the diversity of experiments, as well as for recognizing the idea as a natural yet novel extension. We address the concerns and specific comments below:
> > 1. While a natural idea, the method has limited novelty.
>
> Please refer to the general response.
>
> > 3. All improvements are paid for by significant longer training times per epoch. Therefore, it is unclear how much the use of the OT coupling is really useful here.
>
> We agree that minibatch OT introduces additional computation during training, but we would like to emphasize that this is a one-time cost, and yields substantial inference-time benefits for one-step generation. For point cloud generation (Table 3), OT-MF improves W₂ from 0.0477→0.0121 on ShapeNet Chairs (≈74% reduction) and 0.0377→0.0208 on ModelNet10 (≈45% reduction), while increasing per-epoch training time from 16.32→20.81 s and 23.41→28.64 s (roughly a 27% and 22% overhead), respectively. We believe this moderate increase in training cost is well justified given the gains in transport accuracy.
>
> Moreover, we introduce several OT variants that are more computationally efficient than exact OT yet still provide clear improvements over the original MeanFlow, offering practical alternatives when training efficiency is a priority.

---

### Official Review · Reviewer_khDK · 2025-10-28

**Soundness:** 4
**Presentation:** 4
**Contribution:** 1
**Rating:** 2
**Confidence:** 3

**Summary:**

This paper proposes optimal transport MeanFlow (OT-MF), which integrates OT-based sampling with the MeanFlow framework for one-step generative modeling. The training samples obtained via OT couplings (e.g., Sinkhorn or Linear OT) exhibit geometrically straighter trajectories compared to independent Gaussian-data pairs. This coupling strategy, when combined with the time-averaged MeanFlow formulation, can reduce inference steps while preserving the fidelity of the multi-step flow process. Experiments on diverse modalities (image generation, image-to-image translation, and point cloud generation) demonstrate that proposed method shows improved sample quality, faster inference, and better trajectory alignment compared to vanilla MeanFlow.

**Strengths:**

- The proposed combination of OT and MeanFlow is well-motivated to improve trajectory straightness and sampling efficiency.
- The paper provides comprehensive background and related work on Flow Matching, OT-based methods, and MeanFlow.
- Experiments are conducted on diverse tasks and demonstrate consistent improvements in sample quality and efficiency.
- Incorporation of scalable OT solvers enhances computational overhead without major performance loss.

**Weaknesses:**

- Novelty is weak. The main idea (combining existing OT-based coupling with MeanFlow) is conceptually straightforward. While effective, it primarily extends prior techniques rather than introducing a fundamentally new theoretical framework.
- The proposed method still fails to accurately capture the data distribution, even in a simple 2D toy dataset (Figure 2).
- (Minor comment) The qualitative difference between MF and OT-MF is subtle in Figure 1 illustration. Using multiple (x0, x1) trajectories or average path visualizations could better highlight the ‘trajectory straightening’ effect.

**Questions:**

- The proposed method primarily combines two existing ideas, OT-based flow matching and MeanFlow. Beyond this integration, is there any novel algorithms, theoretical contribution, or architectural enhancement introduced in the paper?
- In Table 1, why does Sinkhorn OT often outperform the proposed OT-MF in terms of Wasserstein-2 distance?
- Is this method still effective in higher-dimensional or sequential generative tasks, such as audio synthesis or text-to-video generation?

---

> ### Author Response · Authors · 2025-12-03
> **Response to Reviewer khDK**
>
> We sincerely thank the reviewer for their careful assessment, positive remarks on the motivation, background discussion, and experimental validation, as well as for the constructive critical feedback. We address the concerns and questions below:
> > 1. The proposed method primarily combines two existing ideas, OT-based flow matching and MeanFlow. Beyond this integration, is there any novel algorithms, theoretical contribution, or architectural enhancement introduced in the paper?
>
> Please refer to the general response.
>
>
> > 2. In Table 1, why does Sinkhorn OT often outperform the proposed OT-MF in terms of Wasserstein-2 distance?
>
> We thank the reviewer for raising this question. This is mainly caused by the entropic regularization $\epsilon = 1/\lambda$ in the Sinkhorn algorithm [1]. This value introduces a trade-off between speed and accuracy. A smaller $\epsilon$ generally introduces better performance in terms of Wasserstein-2 distance, but takes longer to converge. Similarly, as $\epsilon$ increases, we expect the performance of Sinkhorn to drop, and the convergence time to become faster. We show this pattern in one of our toy experiments (Gaussian-to-checkerboard) with a varying value for $\epsilon$.
>
>
>
> | Method  | W-2 @ 1 | TR(ms) |
> | -------- | -------- | -------- |
> | Sinkhorn ($\epsilon$  = 0.1)     | 0.00482     | 40.87     |
> | Sinkhorn ($\epsilon$  = 0.5)     | 0.00664     | 24.32     |
> | Sinkhorn ($\epsilon$  = 1.0)     | 0.01497     | 20.04     |
> | Sinkhorn ($\epsilon$  = 5.0)     | 0.03448     | 16.36     |
> | OT-MF     | 0.00482     | 24.69     |
>
> As demonstrated in the table above, Sinkhorn with a larger $\epsilon$ does not outperform exact OT, but has a more effiecient training time. Please note that the timing results are reported on Google Colab CPU, and thus a slight variation in training time is expected. However, the trade-off pattern between performances of different methods remains the same.
>
>
> > 3. Is this method still effective in higher-dimensional or sequential generative tasks, such as audio synthesis or text-to-video generation?
>
> We thank the reviewer for the thoughtful question Extending OT-MeanFlow to higher-dimensional or sequential domains such as audio or text-to-video can introduce additional challenges (e.g. substantially higher dimensionality) and falls outside the scope of this paper. While the framework is conceptually general, applying it to these modalities would require domain-specific architectures and training setups, which we leave as promising future work.
>
> **References**
>
> [1] Cuturi, M. (2013). Sinkhorn distances: Lightspeed computation of optimal transport. Advances in neural information processing systems, 26.

---

### Official Review · Reviewer_UiSG · 2025-10-29

**Soundness:** 2
**Presentation:** 3
**Contribution:** 2
**Rating:** 2
**Confidence:** 4

**Summary:**

This work proposes integrating the Minibatch OT coupling into Mean Flow training. To ensure the training efficiency, various OT computation methods are adopted, including Sinkhorn, Linear OT, and etc. The proposed method is evaluated on image generation (MNIST), shape-conditioned reconstruction (ShapeNet Chairs and ModelNet10), and image-to-image translation (FFHQ), demonstrating improvements over Mean Flow across these tasks.

**Strengths:**

- Extending Mean Flow with Minibatch OT coupling seems to be an interesting approach to improve one-step generative models by leveraging the demonstrated effectiveness of Minibatch OT.
- Various experiments have shown that the introduction of OT coupling can improve the generation quality.
- The writing and presentation of the manuscript are overall clear and easy to follow.

**Weaknesses:**

However, with regard to the current manuscript, I still have several major concerns:

Limited Novelty: This work primarily combines Mean Flow (Geng et al. 2025) with Minibatch OT (Pooladian et al., 2023; Tong et al., 2023a). Since minibatch OT approaches have been thoroughly explored and proven effective, it is unsurprising that they improve generation quality over Mean Flow. The contribution to the research community appears limited.

That said, a thorough evaluation justifying the necessity of Minibatch OT for Mean Flow could still be a valuable contribution. However, I have several questions about the current evaluation.

Evaluation Tasks: The evaluation tasks do not follow standard protocols. The original Mean Flow paper uses ImageNet 256×256, a standard benchmark for image generation. This work evaluates only on MNIST for image generation, making it unclear whether the conclusions hold at scale. For 3D shape generation, the choice of a reconstruction task (using the input shape as conditioning) is unclear when unconditional generation could be evaluated instead, following the protocol in LION [A].

Evaluation Baselines: The manuscript primarily compares against Mean Flow alone. However, various works have explored techniques for accelerating inference, including adversarial distillation, consistency models, and shortcut models. The current evaluation does not include comparisons with these baselines.

**Questions:**

Overall, I recommend rejection due to limited novelty and insufficient evaluation. The work primarily combines existing techniques (Mean Flow and Minibatch OT) in a straightforward manner, yielding expected rather than surprising improvements. The experimental evaluation uses non-standard benchmarks (MNIST instead of ImageNet 256×256, reconstruction tasks instead of unconditional generation) and lacks comparisons with relevant baselines such as adversarial distillation, consistency models, and shortcut models. A substantially revised submission with comprehensive benchmarking on standard tasks and comparisons against the broader landscape of one-step generation methods would be necessary to assess the true contribution of this work.

Based on these, I have the following questions:
- The evaluation for image generation is limited to MNIST, while the original Mean Flow paper demonstrates results on ImageNet 256×256, which is a standard benchmark in the field. Can you provide experiments on ImageNet 256×256 to demonstrate that OT-MF's improvements generalize to high-resolution, complex image generation tasks? Without this evaluation, it remains unclear whether the benefits of incorporating OT coupling hold at scale.
- The manuscript primarily compares OT-MF against Mean Flow. However, there are several other approaches for one-step or few-step generation, including adversarial distillation methods, consistency models, and shortcut models. Can you include comparisons with these baselines to better contextualize OT-MF's performance relative to the broader landscape of accelerated generative modeling techniques? This would help clarify the practical advantages of your approach.

---

> ### Author Response · Authors · 2025-12-03
> **Response to Reviewer UiSG**
>
> We sincerely thank the reviewer for their positive acknowledgement of our proposed method, results, and writing. We are grateful for the constructive comments, and we summarize our responses to their questions and concerns below:
> > Limited Novelty: This work primarily combines Mean Flow (Geng et al. 2025) with Minibatch OT (Pooladian et al., 2023; Tong et al., 2023a). Since minibatch OT approaches have been thoroughly explored and proven effective, it is unsurprising that they improve generation quality over Mean Flow. The contribution to the research community appears limited.
>
> Please refer to the general response.
>
> > Evaluation Tasks: The evaluation tasks do not follow standard protocols. The original Mean Flow paper uses ImageNet 256×256, a standard benchmark for image generation. This work evaluates only on MNIST for image generation, making it unclear whether the conclusions hold at scale. For 3D shape generation, the choice of a reconstruction task (using the input shape as conditioning) is unclear when unconditional generation could be evaluated instead, following the protocol in LION [A].
>
> Please refer to the general response.
>
> > Evaluation Baselines: The manuscript primarily compares against Mean Flow alone. However, various works have explored techniques for accelerating inference, including adversarial distillation, consistency models, and shortcut models. The current evaluation does not include comparisons with these baselines.
>
> Since our method is a direct extension of MeanFlow, the primary baseline choice was to isolate the effect of integrating OT variants within the MeanFlow framework. However, we agree that comparing our method with other few-step generative approaches will strengthen the findings. In the revised version, we will add comparisons with representative baselines.

---

### Official Review · Reviewer_nBX3 · 2025-11-01

**Soundness:** 2
**Presentation:** 3
**Contribution:** 2
**Rating:** 4
**Confidence:** 4

**Summary:**

The paper integrates optimal transport and mean flows for better one and few steps generation with geometry-aware learning. The author claims the unified framework performs efficiently on few different downstream tasks such as toy example, image generation, point cloud generation and image-to-image translation.

**Strengths:**

1.The paper is well written and easy to follow.
2.The paper leverages OT coupling mechanisms and mean flows on the training objective to enforce globally optimal source–target alignments, leading to straighter and more stable transport trajectories.
3.The work achieved good quality performance in multiple tasks with few step generation.

**Weaknesses:**

1.Limited novelty and incremental improvement: The main contribution lies in combining established OT techniques with mean flows. The paper lacks new theoretical developments or formal analysis to differentiate it from prior OT-regularized flow-matching approaches. The authors are encouraged to clarify what is theoretically or algorithmically new beyond the combination itself.
2.Incomplete baselines of one/few steps generation: The authors achieved better performance compared to meanflow and other OT techniques but the lack of comparison with recent one/few steps SOTA generative models limits the impact of the proposed framework. Models such as consistency/distillation models or few step-diffusion or flow matching baselines are not cited or discussed [1,2,3].
3. Lack of hyperparameter analysis: The paper does not discuss how the hyperparameters (e.g. batch size, learning rate and etc) are selected. The  paper should explain the procedure or tuning strategy of the models prior to the experiments.

References:
[1] Song, Yang, et al. "Consistency models." (2023).
[2] Meng, Chenlin, et al. "On distillation of guided diffusion models." Proceedings of the IEEE/CVF conference on computer vision and pattern recognition. 2023.
[3] Sauer, Axel, et al. "Adversarial diffusion distillation." European Conference on Computer Vision. Cham: Springer Nature Switzerland, 2024.

**Questions:**

note the weakness above

---

> ### Author Response · Authors · 2025-12-03
> **Response to Reviewer nBX3**
>
> We sincerely thank the reviewer for their positive comments on the writing, methodological contributions, and experimental results of our paper. We appreciate their constructive feedback, and we address their questions and concerns below:
> > 1. Limited novelty and incremental improvement: The main contribution lies in combining established OT techniques with mean flows. The paper lacks new theoretical developments or formal analysis to differentiate it from prior OT-regularized flow-matching approaches. The authors are encouraged to clarify what is theoretically or algorithmically new beyond the combination itself.
>
> Please refer to the general response.
>
> > 2. Incomplete baselines of one/few steps generation: The authors achieved better performance compared to meanflow and other OT techniques but the lack of comparison with recent one/few steps SOTA generative models limits the impact of the proposed framework. Models such as consistency/distillation models or few step-diffusion or flow matching baselines are not cited or discussed [1,2,3].
>
> Since our method is built directly on MeanFlow, our primary goal in the current submission is to demonstrate the improvement achieved by incorporating OT variants into the MeanFlow pipeline. However, we agree that comparing our method with broader few-step methods will strengthen the paper. We will revise our paper to expand the experiments and include direct comparisons with representative few-step baselines where feasible.
>
> > 3. Lack of hyperparameter analysis: The paper does not discuss how the hyperparameters (e.g. batch size, learning rate and etc) are selected. The paper should explain the procedure or tuning strategy of the models prior to the experiments.
>
> We thank the reviewer for highlighting this. In our experiments, we have not conducted excessive hyperparameter tuning. To ensure a fair comparison among MeanFlow and all OT-based variants, all models use an identical set of hyperparameters (e.g., batch size, learning rate, optimizer settings) and are trained for the same number of epochs. Since no dataset-specific tuning was performed, we did not include a dedicated tuning section. We will ensure to clarify this in the future revisions.

---

### Author Response · Authors · 2025-12-03
**General Response**

### **Theoretical contributions**

Multiple reviewers have raised concerns on the novelty of our method. We appreciate the emphasis on theoretical novelty, and would like to further clarify the contributions of our work below.

We would like to emphasize that integrating OT with MeanFlow is an unexplored direction with distinct value. Prior OT–flow-matching works (e.g., OT-CFM [1]) operate within multi-step flow trajectories. In contrast, MeanFlow is a single-step generative framework with fundamentally different training. Our work is the first to show that OT can straighten the MeanFlow vector field further, and substantially enhance one-step generative performance across different tasks.


Furthermore, in contrast to prior work, we develop a unified OT-MF framework that supports multiple OT solvers (e.g., Sinkhorn and linear OT) and show empirically that all of them consistently improve MeanFlow across different tasks, while offering different cost–quality trade-offs. We will revise the introduction and method sections to highlight these points and better position our contributions relative to prior OT-regularized flow methods and MeanFlow.

### **Limited experimental scale**

We appreciate the comment raised by multiple reviewers on the scale of our experiments. Our initial experiments focused on MNIST and additional modalities due to time and compute constraints. However, our goal was to also demonstrate that OT-MeanFlow consistently improves MeanFlow across diverse domains, including point cloud generation and image-to-image translation.
We agree that further large‑scale image experiments (e.g., higher resolution) would strengthen the work, and we plan to pursue them as follow‑up.

**References**

[1] Tong, A., Fatras, K., Malkin, N., Huguet, G., Zhang, Y., Rector-Brooks, J., Wolf, G., & Bengio, Y. (2024). Improving and generalizing flow-based generative models with minibatch optimal transport. Transactions on Machine Learning Research, 1-34.

---

### Note · Program_Chairs · 2026-01-17
**Submission Desk Rejected by Program Chairs**

The following references in this submission do not refer to real documents and/or have major errors in bibliographic information:

 Alexander Tong, Yang Liu, et al. Improved techniques for training score-based generative models. In ICML, 2023b.
Yongxin Chen, Tryphon T. Georgiou, and Michele Pavon. Optimal transport under schrödinger bridge regularization. IEEE Transactions on Automatic Control, 66(9):3779-3793, 2021.